# Complete male-to-female sex reversal in XY mice lacking the *miR-17~92* cluster

Alicia Hurtado [1,2,4,5], Irene Mota-Gómez [2,5], Miguel Lao [1,5], Francisca M. Real [3], Johanna Jedamzick[2], Miguel Burgos [1], Darío G. Lupiáñez [2,4,5] ✉, Rafael Jiménez [1,5] ✉ & Francisco J. Barrionuevo [1,5] ✉

Mammalian sex determination is controlled by antagonistic gene cascades operating in embryonic undifferentiated gonads. The expression of the Y-linked gene *SRY* is sufficient to trigger the testicular pathway, whereas its absence in XX embryos leads to ovarian differentiation. Yet, the potential involvement of non-coding regulation in this process remains unclear. Here we show that the deletion of a single microRNA cluster, *miR-17~92*, induces complete primary male-to-female sex reversal in XY mice. *Sry* expression is delayed in XY knockout gonads, which develop as ovaries. Sertoli cell differentiation is reduced, delayed and unable to sustain testicular development. Pre-supporting cells in mutant gonads undergo a transient state of sex ambiguity which is subsequently resolved towards the ovarian fate. The *miR-17~92* predicted target genes are upregulated, affecting the fine regulation of gene networks controlling gonad development. Thus, microRNAs emerge as key components for mammalian sex determination, controlling *Sry* expression timing and Sertoli cell differentiation.

In mammals, phenotypic sex (male or female) depends on the type of gonads developed in the embryo (testes or ovaries), which in turn depends on its sex chromosome endowment (XY or XX). Both testes and ovaries contain three main types of sex-specific cell lineages: 1) germ cells that give rise to gametes, either sperm or oocytes, through meiosis; 2) somatic supporting cells, either Sertoli or granulosa (follicle) cells, that nurse and support the developing germ cells; and 3) somatic steroidogenic cells, either Leydig or theca cells, that produce male and female steroid hormones (either androgens or estrogens). In the embryonic testis, Sertoli and Leydig cells produce hormones (AMH and testosterone, respectively) that masculinise the embryo. In the absence of these hormones, XX embryos develop as phenotypic females. All these cell types differentiate from precursor cell lineages present in the undifferentiated, bipotential gonadal primordium (also called genital ridge) of the embryo, that forms during the organogenesis stages as a thickening of the ventromedial surface of the coelomic epithelium of the mesonephros, which functions as a primitive embryonic kidney (reviewed in ref. 1). Mammalian sex determination involves the simultaneous expression of genes with antagonistic functions[2,3], resulting in a balanced network of opposing ovary- (RSPO1, WNT4, FOXL2, among others) and testis-promoting (SRY, SOX9, FGF9, among others) signalling and transcription factors. The Y-linked gene, *SRY*, is the trigger that breaks this balance: its expression in the pre-supporting cell lineage of the undifferentiated gonads from XY individuals induces testis differentiation, whereas its absence in XX gonads results in ovarian development[4–6]. In mice, at the time of *Sry* expression in XY supporting cell precursors, these cells become pre-Sertoli cells, which will thus differentiate as embryonic

[1]Department of Genetics and Institute of Biotechnology, Labs. 127 and A105, Centre for Biomedical Research, University of Granada, Armilla, Granada, Spain. [2]Epigenetics and Sex Development Group, Berlin Institute for Medical Systems Biology, Max-Delbrück Center for Molecular Medicine, Berlin, Germany. [3]Research Group Development & Disease, Max Planck Institute for Molecular Genetics, Berlin, Germany. [4]Present address: Centro Andaluz de Biología del Desarrollo (CABD), CSIC/UPO/JA, Seville, Spain. [5]These authors contributed equally: Alicia Hurtado, Irene Mota-Gómez, Miguel Lao, Darío G. Lupiáñez, Rafael Jiménez, Francisco J. Barrionuevo. ✉e-mail: dario.lupianez@csic.es; rjimenez@ugr.es; fjbarrio@go.ugr.es

Sertoli cells. At the same time, XX counterparts are specified as pre-granulosa cells, which differentiate as granulosa cells. In this bipotential system, the repression of genes promoting the alternate fate is paramount for sex-specific differentiation[1,7]. Further, this mutual antagonism is considered as a landmark of both mammalian sex determination and maintenance, highlighting the strong genetic component that influences this developmental process.

Previous research in the sex determination field has mostly focused on the identification of genes controlling this process and on elucidating their hierarchical relationships at a molecular level. Yet, the involvement of non-coding regulatory mechanisms in mammalian sex determination remains largely unexplored[8,9]. One intriguing class of non-coding regulators are microRNAs (miRNAs), short nucleotide sequences that mediate the post-transcriptional regulation of gene expression. These small molecules were initially characterised as temporal regulators of cell-fate decisions[10–12]: their inactivation led to heterochronic shifts in gene expression and differentiation. Progressively, miRNAs have been also shown to be implicated in numerous developmental and pathological processes[13]. During sex development, both miRNAs and their associated machinery are differentially regulated in developing testes and ovaries[14]. Furthermore, the disruption of their biosynthesis in adult mice compromised their fertility, confirming a role of miRNAs in maintaining gonadal cell type identities[15]. However, the inactivation of *Dicer1*, an essential component for miRNA biosynthesis, prior to the sex determination stage did not affect this process[16], but the long-time persistence of miRNAs upon disruption of their biosynthesis machinery and/or the inefficient *Sf1Cre*-induced recombination in coelomic epithelial cells at E11.5[17] may have obscured their potential involvement in sex determination.

One of the most well-studied microRNAs is the *miR-17-92* cluster. This polycistronic cluster, also called OncomiR-1, contains six miRNA genes (*miR-17, miR-18a, miR-19a, miR-20a, miR-19b-1* and *miR-92a-1*; Supplementary Fig. 1a), known to be involved in a wide range of developmental and pathogenic processes[18]. In the adult testis, *miR-17-92* maintains transcriptomic levels within normal values[19,20] and, in cooperation with its paralog cluster *miR-106b-25*, it is required to sustain male fertility[21]. Interestingly, *miR-17-92* cluster members are expressed in mouse gonads from both sexes during and after the sex determination stage (Supplementary Fig. 1b) and their predicted targets include numerous sex-related genes (see below). Also, miRNAs of this cluster are evolutionarily well conserved and are related to sex development and sex change in non-mammalian vertebrates[22]. However, a possible role of the miR–17-92 cluster in mammalian sex determination has not been previously investigated.

Here we demonstrate that the ablation of the *miR-17-92* cluster, prior to sex determination, causes complete male-to-female sex reversal in mice. By combining bulk and single-cell RNA-seq (scRNA-seq) and time-course expression analyses, we demonstrate that *Sry* expression is delayed in the absence of *miR-17-92*. As a consequence of this heterochrony, pre-supporting cells undergo a transient state of sexual ambiguity that likely causes failed differentiation into Sertoli cells and the subsequent activation of the ovarian program. Furthermore, we show that *miR-17-92* ablation results in a misregulation of hundreds of target genes, simultaneously affecting the regulation of critical gene networks. Our findings reveal an unexpected role for miRNAs in controlling the timing of *Sry* expression and Sertoli cell differentiation.

## Results

### XY embryos lacking *miR-17-92* develop as phenotypic females

To investigate the potential role of the *miR-17-92* cluster in mammalian sex determination, we analysed the genito-urinary system of knockout (KO) mice, generated by using the Cre/LoxP system (*Tg(CAG-cre)^INagy;miR-17-92^del/del*). As *miR-17-92* KO mice undergo neonatal lethality[23], we focused our analysis on the embryonic day 17.5 (E17.5),

when sex is already differentiated. While XY control foetuses displayed normal descended testes, XY mutants developed as phenotypic females with two uterine horns and ovaries directly inferolateral to the kidneys, similar to those observed in both XX mutants and control females (Fig. 1a–d). Histological analyses on XY KO gonads (Fig. 1f) revealed a clear ovarian morphology, indistinguishable from that of XX mutants or control ovaries (Fig. 1g, h). Testis-specific structures, such as testicular cords or large blood vessels like those observed in XY control testes (Fig. 1e) are completely absent in XY mutant gonads. Instead, gonadal structure was completely indistinguishable from XX mutants or control ovaries (Fig. 1e–h). Double immunofluorescence showed that the testicular marker SOX9 was exclusively present in XY control gonads, whereas XY mutant gonads, as XX gonads, only expressed the ovarian marker FOXL2 (Fig. 1i–l). Consistently, the activation of germ cell meiosis, an early sign of ovarian development, was observed in XY mutant and XX gonads, but not in XY controls (Fig. 1m–p). Altogether, these results show that, in the absence of the *miR-17-92* cluster, XY mice undergo complete primary sex reversal.

### *miR-17-92* regulates gonadal growth before sex determination

To explore the molecular mechanisms associated to sex reversal in *miR-17-92* mutants, we performed bulk RNA-seq on gonads during the sex determination period and shortly afterwards (E11.5 and E12.5). These data permitted us to verify the effective deletion of the *miR-17-92* cluster by using the frequencies of gonadal transcript reads mapped to the *Mir17hg* locus, where the *miR-17-92* cluster is located (Fig. 2a; Supplementary Fig. 2). Pairwise comparisons between the four analysed genotypes at the two developmental stages identified a variable number of differentially expressing genes (DEGs; Supplementary Data 1–9). The DEGs between XX and XY control gonads at E12.5 (3569 DEGs; FDR < 0.05) were used as a reference set for genes with normal dimorphic expression. Log$_2$-fold-change heat maps at E12.5 revealed that both XY and XX mutants acquire a female-like expression profile, with most genes displaying an ovarian-specific expression pattern (Fig. 2b). Consistent with this observation, multi-dimensional scale plot revealed moderate differences between mutants and controls at E11.5, with mutant conditions clustering closer together. At E12.5 these differences were more prominent, with XY control replicates clustering separately from all other samples (Supplementary Fig. 3a), indicating that mutant gonads follow the ovarian fate shortly after the sex determination period.

We observed that the number of DEGs between XX and XY mutant gonads was marginal in both E11.5 and E12.5 stages (12 and 21 DEGs, respectively; Supplementary Data 1–9), denoting the existence of *miR-17-92*-related signatures that are shared between sexes. Furthermore, Gene Ontology analysis identified several enriched GO terms when E11.5 mutants were compared to controls in both XX and XY backgrounds (Supplementary Fig. 3b; Supplementary Data 10 and 11). These observations suggest that *miR-17-92* may control some non-sex related aspects of gonadal development. In particular, we found GO terms related to cell proliferation and molecular pathways involved in its control (Ras protein signal transduction, regulation of MAP kinase activity, ERK1 and ERK2 cascade, among others). Consistently, *miR-17-92* KO gonads were smaller than their wildtype counterparts at E11.5, displaying a mean volume of ~50% compared to controls (Fig. 2c). In contrast, as previously described[23], the body size of mutant and control embryos was similar until E13.5, when the growth of mutant embryos was retarded compared to controls. Particularly, at the sex determination stage (E11.5) the gross morphology and size of mutant embryos was similar to that of controls (Supplementary Fig. 3c), indicating a gonad-specific proliferation defect. Based on these results, we investigated proliferation rates in developing gonads using double immunofluorescence against the Wilms tumour protein (WT1), a progenitor cell marker, as well as the proliferation marker Ki67. This analysis revealed that the genital ridge of mutant gonads contained

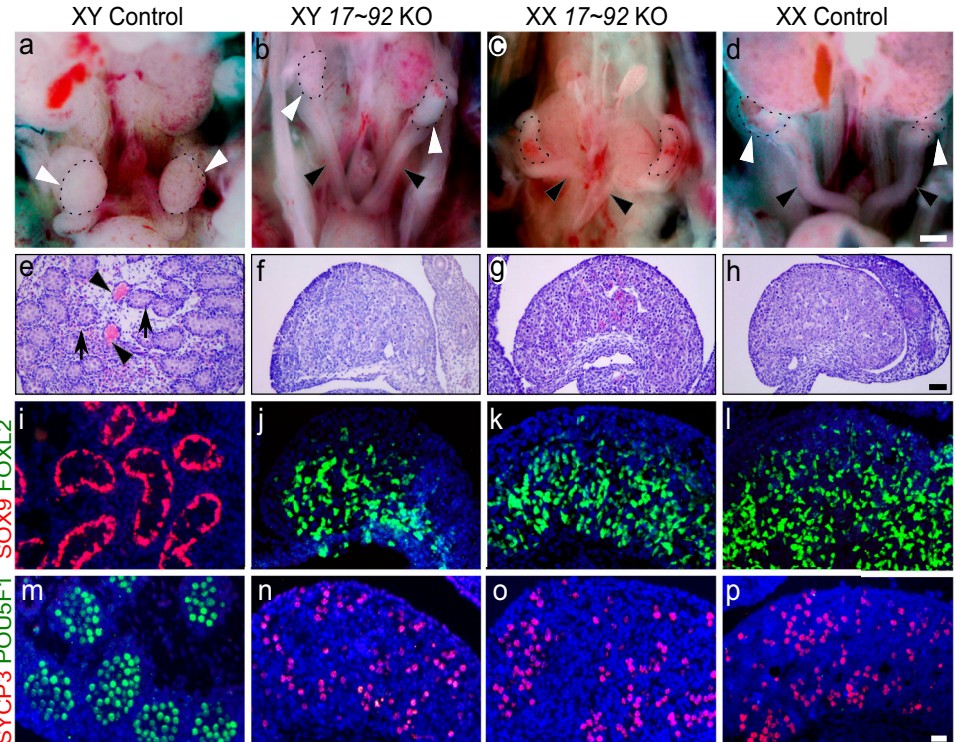

**Fig. 1 | Male-to-female sex reversal in *miR-17~92* KO mice. a–d** Urogenital anatomy of control and mutant XY and XX embryos at E17.5. Gonads are encircled by a dotted line and marked by white arrowheads. Uterine cornua are marked with black arrowheads. Note that only control XY embryos develop testes, while the rest develop ovaries and uterine cornua. **e–h** Gonadal histology at E17.5. Control testes contained solid cords (arrows) and large blood vessels (arrowheads) (**e**). In contrast, XY mutant gonads (**f**) showed a typical ovarian histology lacking both cords and vessels, as observed in XX gonads (**g**, **h**). Double immunofluorescence at E17.5 showing that only XY control testis expressed the testicular marker SOX9 (**i**), while the other three genotypes expressed the ovarian marker FOXL2 (**j–l**).

**m–p** Double immunofluorescence for the pluripotency marker POU5F1 (OCT4) and the meiotic prophase cell marker SYCP3 at E17.5. Only germ cells in XY control testes expressed POU5F1 and showed no SYCP3 expression, evidencing the absence of meiotic cells in embryonic testes (**m**). Contrarily, all germ cells of the other three genotypes had entered meiosis, as they expressed SYCP3, evidencing full ovarian development (**n**, **p**). Similar results were obtained for the 3–6 animals of each genotype we analysed. Scale bars shown in **d**, **h**, and **p** represent 500 μm for **a–d**, 100 μm for **e–h**; and 20 μm for **i–p**. *miR-17-92* KO mice were generated using the Cre/LoxP system.

less proliferating progenitor cells (WT1⁺, Ki67⁺) than that of controls, before (E11.0; 14–16 tail somites, ts) but not at the sex determination stage (E11.5; 18–19 ts; Fig. 2d). In addition, we found no differences in apoptosis in any of the stages analysed (Supplementary Fig. 3d), with the number of apoptotic cells being very low in all cases. Altogether, these results indicate that *miR-17-92* regulates molecular pathways necessary for the normal growth of the bipotential genital ridges before the sex determination stage.

### *miR-17-92* is required for proper *Sry* expression timing and Sertoli cell differentiation

The reduced proliferation in early *miR-17-92* KO gonads, which is consistent with their overall reduced size, may alter the relative proportions of gonadal cell types. Therefore, we explored changes in cellular composition and identity in *miR-17-92* KO gonads, by performing single-cell RNA-seq (scRNA-seq) experiments. Cre-Lox breeding would yield a low number of embryos with the appropriate genotype (see also Methods section for an extended description), making scRNA-seq experiments technically difficult. Therefore, we employed CRISPR/Cas9 to generate a homozygous deletion of the miRNA cluster in mouse embryonic stem cells (mESC) and subsequently derived mutant animals via tetraploid complementation assays[24,25] (Supplementary Fig. 4a). Of note, these mutants displayed an identical phenotype to those previously generated via Cre-LoxP breeding (Supplementary Fig. 4b). We analysed the mesonephros/gonad tissue from XY *miR-17-92* KO embryos, as well as of XY and XX controls at E11.5. The profiled cells were clustered based on their

transcriptional similarities, recapitulating all major cell populations previously described in developing gonads[26] (Fig. 3a, b and Supplementary Fig. 5a, b). All cell clusters were present in mutant and control samples, irrespectively of the sex, with the only exception of Sertoli cells, a male-specific cell type that only appeared in XY control gonads (Fig. 3c, d). To elucidate if the lack of Sertoli cells in XY *miR-17-92* mutant gonads was due to delayed or impaired testis differentiation, we also analysed an additional sample of XY *miR-17-92* KO gonads at a later stage (E11.75). Cluster integration confirmed that Sertoli cells were also present in *miR-17-92* mutants, albeit their number was lower than in XY control gonads at E11.5 (5.52% and 10.37%, respectively; Fig. 3b, d). These results indicate that *miR-17-92* is required for proper extent and timing of Sertoli cell differentiation.

To investigate the mechanisms of *miR-17-92*-mediated gene expression, we examined the DEGs between XY control and mutant gonads (Supplementary Data 12–18). Although miRNAs mostly control protein production, previous studies have shown that the changes in protein levels induced by miRNAs are largely mirrored at the transcript level[27–29]. Since the individual components of the *miR-17-92* cluster can be grouped into four "seed families", based on their main target sequences[18] (Supplementary Fig. 1a), we generated cumulative distribution fraction (CDF) plots of the DEG Log$_2$FCs. In both bulk and scRNAseq datasets, the predicted target genes of the four seed families were preferentially upregulated in mutant gonads, compared to non-target genes (Supplementary Fig. 5c), indicating that they are negatively regulated by *miR-17-92* during normal testis development. To visualise the relative contribution of each seed family to gene

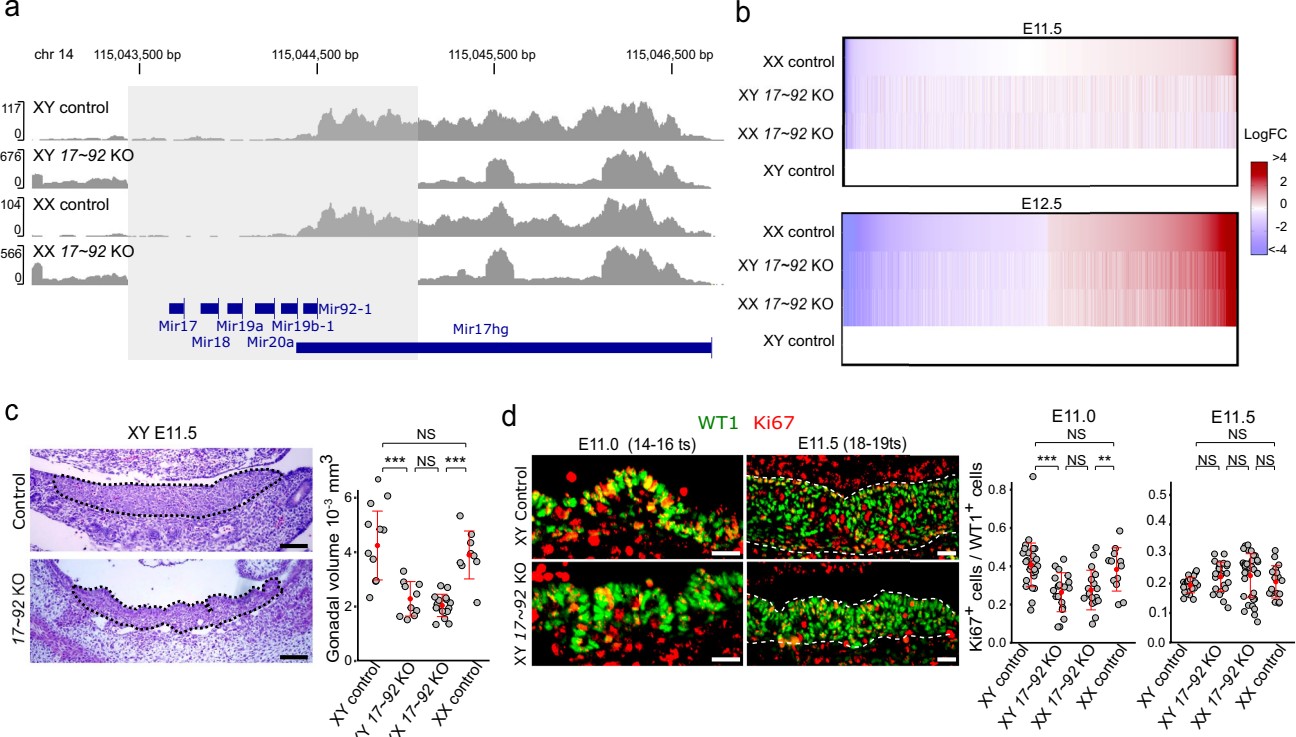

**Fig. 2 | Bulk RNA-seq and morphometric analysis of *miR-17-92* KO gonads during sex determination. a** Representation of the frequencies of gonadal transcript reads mapped to the *Mir17hg* locus. In contrast to control gonads, no reads are observed in the *miR-17-92* region (area highlighted in light grey) of the *miR-17-92* KO gonads. **b** Log₂-fold-change heat maps of DEGs detected between gonads of four different genotypes. Note the similarities at E11.5, as well as the female-like RNA profile of E12.5 mutants, irrespectively of their sex. **c** Haematoxylin and Eosin staining of gonadal sections. Mutant gonads show an irregular contour, including deep transversal furrows, and are thinner and shorter than controls, with a volume reduction of 50% at E11.5 (18–19 ts). For statistical analysis, a two-tail Student's *t* test was performed (*n* = 3); XYcontrol vs XXcontrol, *p* = 5.014330e-01; XYcontrol vs XY17-92KO, *p* = 3.622441e−04; XXcontrol vs XX17-92KO, *p* = 2.356458e−07; XY17-92KO vs XX17-92KO, *p* = 2.462459e−01. **d** Cell proliferation analysis using double immunofluorescence for the gonadal progenitor cell marker WT1 and the cell proliferation marker, Ki67, in gonads both before (E11.0; 14–16 ts) and at the sex determination stage (E11.5; 18–19 ts). Quantification of the mitotic index (number of Ki67⁺ cells/number of WT1⁺ cells) indicate reduced cell proliferation in XX and XY mutant gonads compared to controls before (E11.0; 14–16 ts) but not at the sex determination stage (E11.5; 18–19 ts). For statistical analysis, a Fisher test with Bonferroni adjust was performed. For E11.0: XYcontrol (*n* = 3) vs XXcontrol (*n* = 4), *p* = 1; XYcontrol vs XY17-92KO (*n* = 3), *p* = 1.1e−05; XXcontrol vs XX17-92KO (*n* = 3), *p* = 0.0070; XY17-92KO vs XX17-92KO, *P* = 1; for E11.5: XYcontrol (*n* = 4) vs XXcontrol (*n* = 2), *p* = 1; XYcontrol vs XY17-92KO (*n* = 3), *p* = 0.194; XXcontrol vs XX17-92KO (*n* = 4), *p* = 0.701; XY17-92KO vs XX17-92KO, *p* = 1. Regarding micrographs in panels (**c**) and (**d**), similar results were obtained for the 5 animals of each genotype we analysed. Quantitative data are presented as mean values ± SD. *\**p* < 0.05, *\*\**p* < 0.01, *\*\*\**p* < 0.001. Scale bars in (**c**) and (**d**) represent 50 and 20 μm, respectively. Chr 14 chromosome 14. Mutant mice were generated using the Cre/LoxP system.

regulation, we generated circos plots in which the direction and amplitude of deregulation, as well as the presence of predicted binding sites, was depicted (Fig. 4a; Supplementary Fig. 6). As previously noted, the predicted target genes of the four seed families were preferentially upregulated in the XY mutant gonads compared to XY controls. However, we observed that many non-putative target genes of the cluster were also deregulated, although in this case, there was no preference in the direction of deregulation. Furthermore, the changes in the expression levels of the deregulated genes were generally modest (less than two times; Fig. 4a and Supplementary Data 12–18). These results are consistent with previous studies performed in tail bud and heart from mouse mutant embryos, which showed that *miR-17-92* members act as fine-tuners of large gene networks rather than regulating the expression of a few genes coding for specific transcription factors[30].

Since Sertoli cells do not differentiate properly in XY mutant gonads, we explored the effects of *miR-17-92* ablation on their progenitors, the pre-supporting cells. Gene Ontology analysis on DEGs from XY control and mutant pre-supporting cells revealed an enrichment in categories associated with known biological process, ("cell junction assembly"[31], "precursor metabolite energy"[32], and "reproductive system development") and signalling pathways (MAPK[33], WNT[7,34]) operating in gonadal supporting cell differentiation.

(Supplementary Fig. 7a; Supplementary Data 19). Gene concept analysis also revealed a large network in which different sub-networks shared numerous genes and *miR-17-92* predicted targets (Fig. 4b). In general, the percentage of upregulated *miR-17-92* target genes in XY mutant pre-supporting cells for each GO category ranged from 30% to 40%, whereas downregulated target genes where normally below 7% (Supplementary Fig. 7b). Again, these results suggest that the *miR-17-92* cluster regulates the expression of numerous target genes that may, in turn, fine-tune the expression of large gene networks during sex determination. Among these gene networks we found "sex differentiation" and "Wnt signalling pathway", the latter being essential for ovary differentiation[7,34]. In particular, DEGs belonging to the WNT pathway were preferentially upregulated in XY mutant pre-supporting cells (52 genes upregulated and 7 downregulated; Fig. 4c), and 40% of these genes were putative targets of the *miR-17-92* cluster (Fig. 4c; Supplementary Fig. 7b). In addition, *Foxl2* and *Fst*, two key factors during ovarian development[35,36], which are also *miR-17-92* predicted targets, were upregulated in XY mutant gonads (Fig. 4c; Supplementary Data 14). These results suggest a function for the *miR-17-92* cluster in maintaining female-promoting genes and signalling pathways downregulated during early testis differentiation.

However, we also found that *Sox9*, which encodes a transcription factor essential for testis development[37,38] but is not a putative

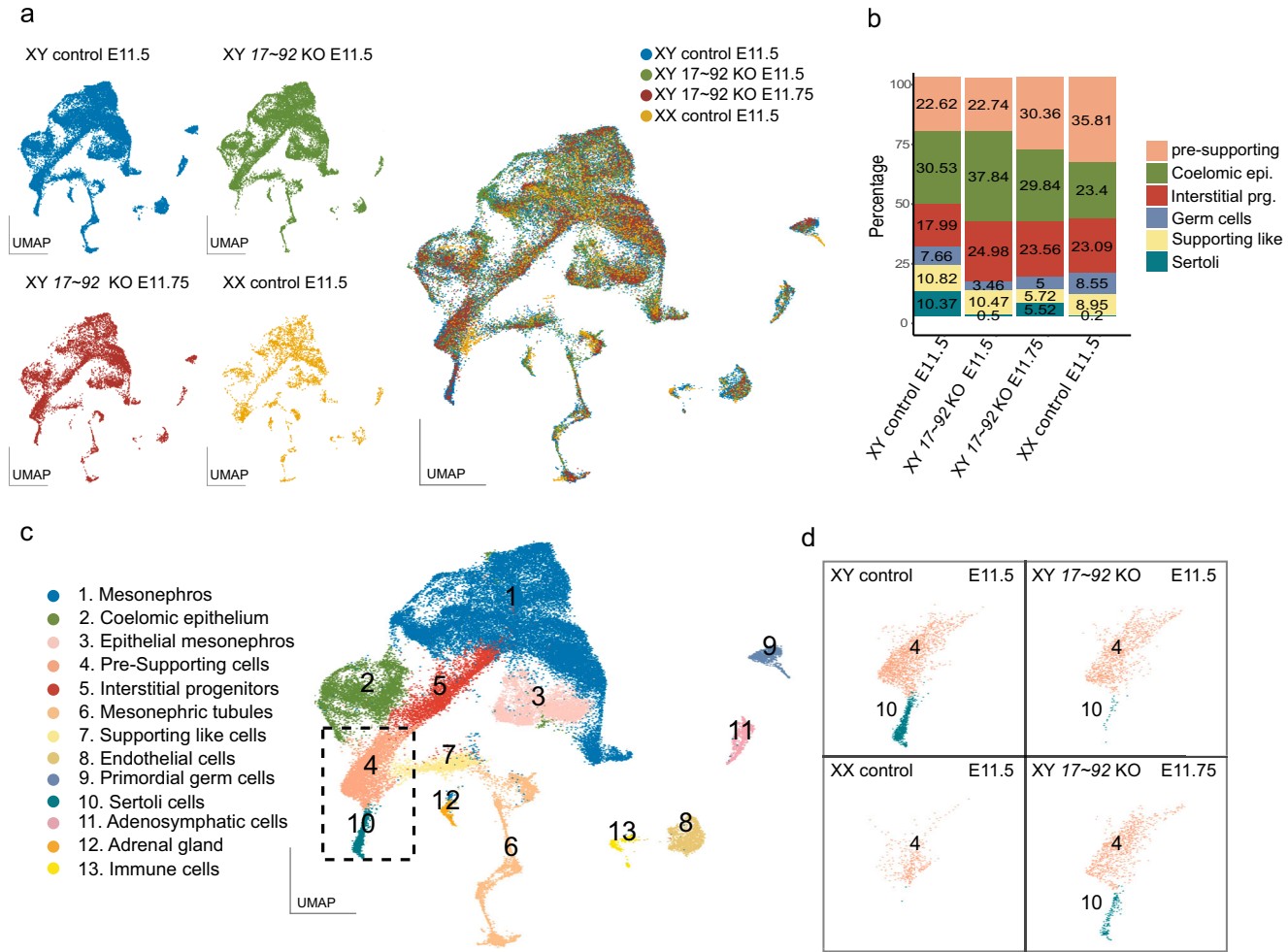

**Fig. 3 | Single cell RNA-seq analysis of *miR-17~92* KO gonads during sex determination.** For clustering and identification of populations, cells were coloured by genotypes (**a**) and by cell clusters (**c**). All major gonadal cell populations were present in both mutant and control gonads, except Sertoli cells that appeared only in XY controls at E11.5. Note that these cells were identified in XY mutants at a later timepoint E11.75 (**b** and **d**). UMAP, uniform manifold approximation and projection. Data from gonads of *miR-17-92* KO mice generated by CRISPR/Cas-tetraploid aggregation.

*miR-17-92* target, was downregulated in mutant pre-supporting cells (Fig. 4c; Supplementary Data 12). Therefore, we investigated the expression of the two key factors necessary for Sertoli cell differentiation in mammals: SOX9 itself, and its main activator, the testis-determining factor SRY. Immunofluorescence analyses revealed that cells expressing the SRY protein were already present by E11.25 in XY control testes (16–17 ts), with their number peaking at E11.5-E11.75 (~18–21 ts) and declining afterwards until almost disappearing by E12.5 (28–30 ts). In contrast, the first SRY⁺ cells in XY mutant gonads were detected at E11.5 (18–19 ts), and this number progressively increased until E12.5 (28–30 ts). Consistently, *Sry* transcript levels in mutant XY gonads were significantly lower than those of XY controls at E11.5, whereas the opposite situation was observed at E12.5. (Fig. 4d; Supplementary Data 3 and 7). As previously described[39], the number of SOX9⁺ cells increased dramatically in E11.5 control testes and remained elevated at subsequent stages. In XY mutants, however, SOX9 was not detected at E11.5 (18–19 ts), and SOX9⁺ cells were notably less abundant than in XY controls at subsequently stages (20–21ts and onwards), as shown also at the transcript level (Fig. 4e). It is known that *Sry* must act within a critical time window during mouse sex determination stage to induce testis differentiation[40,41]. Thus, the temporal delay of *Sry* expression and the reduced number of SRY-expressing cells provides a mechanistic basis for the deficient *Sox9* upregulation and Sertoli cell differentiation of XY *miR-17-92* KO gonads.

## A transient state of sexual ambiguity in *miR-17~92* KO pre-supporting cells

We next investigated granulosa cell differentiation by immunofluorescence for the ovarian marker FOXL2. In XY mutant gonads, we observed that FOXL2 is activated shortly after the sex determination window (E11.75; ~20–21 ts), displaying a sustained expression throughout all subsequent stages (Supplementary Fig. 8). This expression onset coincides with that observed in XX control gonads and occurs after the initiation of *Sry* expression (Fig. 5a). This profile is consistent with the hypothesis that granulosa cell differentiation in XY mutant gonads results from delayed *Sry* and *Sox9* expression, although a precocious activation of female-promoting genes carrying *miR-17-92* binding sites cannot be completely ruled out. The analysis of gonads at later stages showed that the vast majority of XY mutant cells expressing FOXL2 did also express SRY (Fig. 5a), revealing that *Foxl2* was preferentially upregulated in pre-supporting cells. Consistently, FOXL2-SOX9 double immunofluorescence showed that, at E12.0 (22–23ts), most of FOXL2⁺ cells coexpress also SOX9 (Fig. 5b). Nevertheless, XY *miR-17-92* KO gonads also contained many FOXL2⁺ SOX9⁻ cells, suggesting that many pre-granulosa cells in XY *miR-17-92* KO gonads differentiate directly from undifferentiated pre-supporting cells. These results indicate that mutant pre-supporting cells undergo a transient state of sex ambiguity, characterised by the concomitant initiation of both the female and the male pathways.

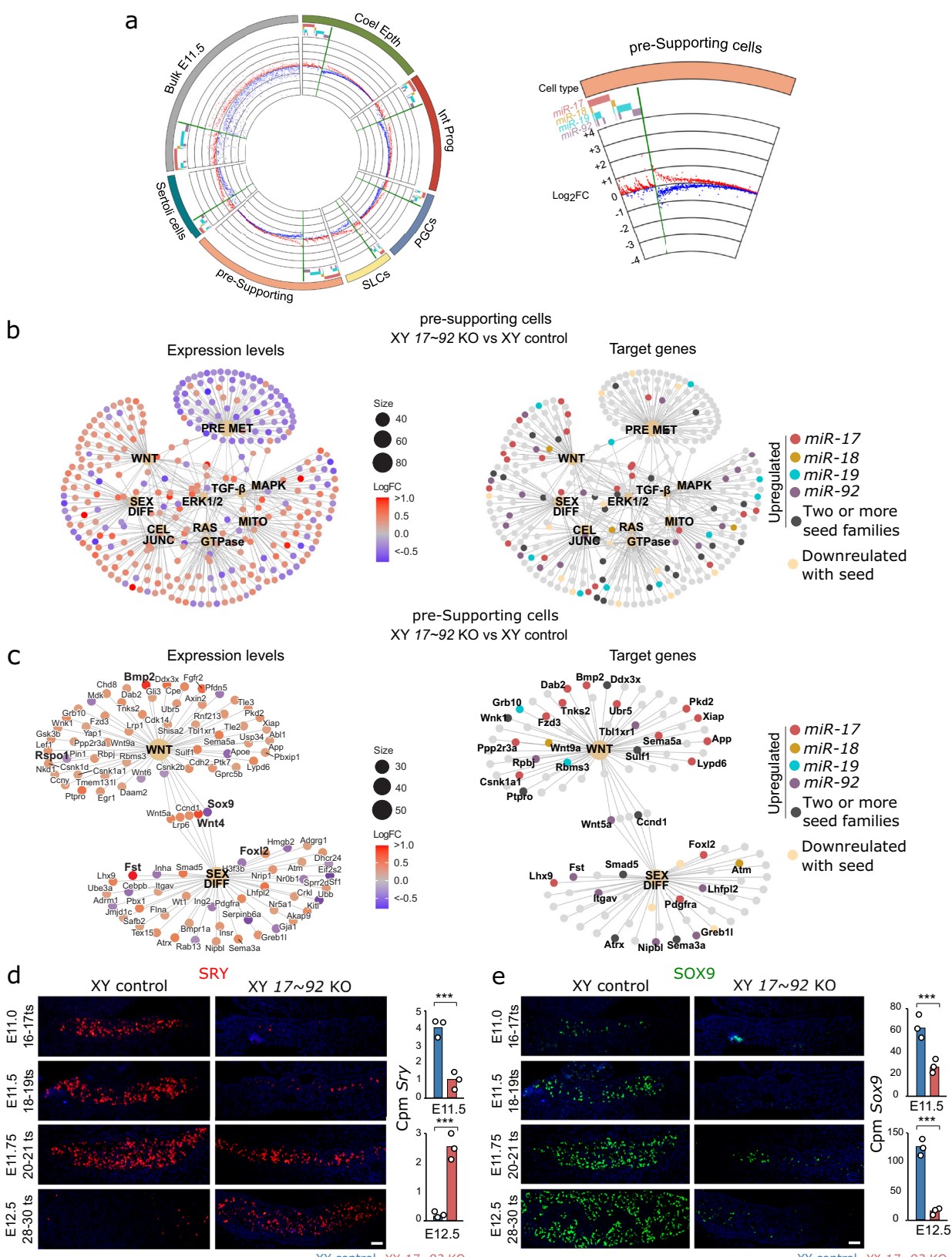

In contrast to E12.0, most FOXL2+ cells were SOX9- at E12.5 (Fig. 5b). In addition to *Foxl2*, other female-specific markers, including *Wnt4*, *Rspo1*, *Bmp2* and *Fst* were upregulated in XY mutant gonads at E12.5, whereas male-specific markers such as *Ptgds*, *Sox8*, *Amh*, and *Dhh* were downregulated (Supplementary Fig. 9a; Supplementary Data 3 and 7). This indicates that the dual sexual program observed in XY mutant gonads shortly after sex determination is resolved at later stages in favour of the female pathway. At E14.5, only a few scattered SOX9+ cells remain in XY mutant gonads, which showed a normal ovarian development including the presence of FOXL2+ supporting cells, meiotic germ cells and a lack of male-specific cell populations or structures, evidencing complete

**Fig. 4 | Effects of *miR-17-92* cluster depletion on gene expression. a** Circos plot of DEGs in cell populations of XY *miR-17-92* KO gonads (see Supplementary Fig. 6 for a high resolution image). Left panel, genes with predicted binding sites for *miR-17-92* seed families (target genes) are grouped at the left side of each sector and marked with a coloured bar in the outer circle (separated by a green bar). A scatter plot of $\log_2$ FC is shown in the inner circle, with up- and downregulated genes labelled in red and blue, respectively. Right panel, magnification of the pre-supporting cells sector. Non-target genes (on the right of the green bar) do not show any particular bias towards up- or downregulation, whereas *miR-17-92* target genes (on the left) are preferentially upregulated. **b** Gene-concept networks using DEGs between XY mutant and control pre-supporting cells at E11.5-E11.75. $\log_2$FCs (left) and predicted targets (right) are depicted. Deregulated functions and pathways are shown. **c** Gene-concept network of DEGs involved in sex determination. Most WNT pathway genes are downregulated and important genes in sex determination, such as *Sox9*, *Foxl2* and *Wnt4* are deregulated. **d** Immunofluorescence for

SRY during sex determination. Note that SRY expression is delayed in XY mutant gonads, compared to XY controls. *Sry* expression levels in the bulk RNA-seq analysis are shown on the right panel. **e** Immunofluorescence for SOX9 during sex determination. Note that the onset of SOX9 expression is delayed in mutant gonads (it starts at 20–21 ts), and the number of SOX9$^+$ cells is reduced. *Sox9* expression levels are also shown on the right panel. Regarding micrographs in panels d and e, similar results were obtained for the 3 animals of each genotype and developmental stage we analysed. For statistical analysis we used the glmQLFit function of the bio-conductor EdgeR package for multiple comparisons ($n = 3$). *Sry*: E11.5, FDR = 0.0049; E12.5, FDR = 0.0012; *Sox9*: E11.5, FDR = 0.0009; E12.5, FDR = 0.0000. Coel. Epth. Coelomic Epithelium, Int. Prog. Interstitial Progenitors, PGCs Primordial Germ Cells, SLCs supporting-like cells. Scale bars shown in (**d**) and (**e**) represent 50 μm. ***$p < 0.001$. Mutant mice in (**a**–**c**) were generated using the CRISPR/Cas-tetraploid aggregation method whereas those in (**d**) and (**e**) were produced using the Cre/LoxP system.

male-to-female primary sex reversal (Fig. 5c; Supplementary Fig. 9b).

Finally, since the mouse *Foxl2* gene has a validated *miR-17* binding site on its 3′UTR region and overexpression of *Foxl2* induces male-to-female sex reversal in transgenic mice, we induced the ablation of the *miR-17* binding site in the *Foxl2* 3′UTR, showing that no sex differentiation effect was produced, probably because the ablation of a single miRNA target site in a single gene is not enough to affect the female pathway significantly (Supplementary note; Supplementary Fig. 10).

## Discussion

Since miRNAs discovery, numerous attempts have been made to elucidate their role during mammalian sex determination. However, to date, none of them could demonstrate that such post-transcriptional regulators can play a fundamental role in this process[8,9]. Indeed, previous studies compromising the biogenesis of these molecules in mouse gonads suggested that they are dispensable for sex determination[15,16]. Contrarily, here we show that a single cluster of miRNAs is essential for mouse testis differentiation. Our results reveal that *miR-17-92* controls two main events of mouse sex determination, namely early gonadal growth and *Sry* expression dynamics. Proliferative growth of the bipotential gonadal primordia is essential to establish the pool of gonadal somatic progenitors, including pre-Sertoli cells, whose number must reach a minimum threshold prior to sex determination to ensure subsequent testis differentiation[42,43]. The gonads of both XX and XY *miR-17-92* KO embryos are smaller than those of control mice (50% reduction) at the sex determination stage, which is a consequence of reduced proliferation during the bipotential stage. The fact that the gonads of mutant embryos are clearly smaller than those of controls, even with similar body size at the time of sex determination, is evidence for a specific effect on the gonads of *mir17-92* ablation, and not for a pleiotropic effect on overall body growth. This proliferative reduction in mutant gonads appears to affect all progenitor cell types similarly, as their relative numbers do not differ significantly from those of controls, notably in the case of the pre-supporting cell lineage (Fig. 3b). Hence, the number of pre-Sertoli cells present at the time of sex determination (E11.5), although reduced, should be sufficient to initiate testis differentiation[43]. Despite of this, *miR-17-92* mutant gonads contain almost no Sertoli cells at E11.5, and shortly later (E11.75) their number is only half that of control testes, a fact that compromises testicular but not ovarian differentiation. The lack of Sertoli cells in XY *miR-17-92* mutant gonads at the sex determination stage is probably a direct consequence of the heterochronic (delayed) expression of *Sry*, as it is well known that the mouse *Sry* gene must act within a critical, narrow time window (between E11.0 and E11.25; 6–8 h) to activate *Sox9* and trigger the male pathway[40,41]. *Sry* expression in XY *miR-17-92* mutants is delayed by about 12 h,

which explains the lack of *Sox9* up-regulation during sex determination and the subsequent XY sex reversal. Interestingly, miRNA knockouts have been repeatedly reported to cause heterochronic shifts in developmental processes[10–12] across a wide range of animal models. In fact, the miRNAs of the *miR-17-92* cluster are evolutionarily well conserved, appearing around 500 Mya in primitive verte-brates, where they have been shown to regulate genes involved in gonadal development and sex change[22]. Overall, this suggests the existence of mechanisms for mRNA function that might be commonly shared across metazoan evolution.

The delayed *Sry* expression is likely indirectly derived from the lack of *miR-17-92*, as *Sry* itself is not a predicted target of these miRNAs. Our results also show that *miR-17-92* predicted targets were preferentially upregulated in XY mutant pre-supporting cells during sex determination, with a misregulation of large gene networks involved in *Sry* regulation and Sertoli cell differentiation. One of these networks is the MAPK signalling pathway, which is involved in the control of both gonadal cell proliferation and *Sry* expression timing. In fact, similar cases of XY sex reversal were described in mice with disrupted MAPK signalling[33,44,45]. We found that 27 genes of the MAPK pathway were deregulated in XY *miR-17-92* KO pre-supporting cells, 40% of them being upregulated target genes (Supplementary Figs. 7a, b and 11a). Interestingly, *Gadd45γ*, which is required for pre-Sertoli cell fate specification in vivo by promoting p38 MAPK signalling[45], shows reduced expression in mutant gonads (Supplementary Fig. 11a, b). In addition, *miR-17-92* regulates gene networks involved in ovarian differentiation in gonadal pre-supporting cells (Fig. 4b). One of these is the WNT signalling pathway, which antagonizes Sertoli cell differentiation and is upregulated in *miR-17-92* XY mutants (Fig. 4c), as expected in gonads that eventually develop as ovaries. Upregulation of the WNT signalling pathway is probably an indirect effect of the inactivation of the male genetic program, but its modest amplitude at E11.5 and the high number of *miR-17-92* putative targets genes that are upregulated in this pathway (40%) also argue for a complementary direct effect of *miR-17-92* depletion, at least during the initial stages of sex determination. Hence, an additional function of *miR-17-92* could be to modestly negatively regulate the WNT signalling pathway at the time of sex determination, an effect that favours testis differentiation but does not affect ovarian development. Another biological process affected in XY mutant cells is "generation of precursor metabolites and energy", which is downregulated in XY mutant pre-supporting cells (Fig. 4b and Supplementary Fig. 7). Interestingly, it was previously shown that a high-glucose metabolic state is required in developing pre-supporting cells for the establishment of SOX9 expression and testis differentiation[32].

Taken together, our results show that *miR-17-92* acts as a cross-cutting regulator that modulates the expression of genes networks and signalling pathways that control processes including proliferative growth, energy metabolism, and female pathway antagonism. As such,

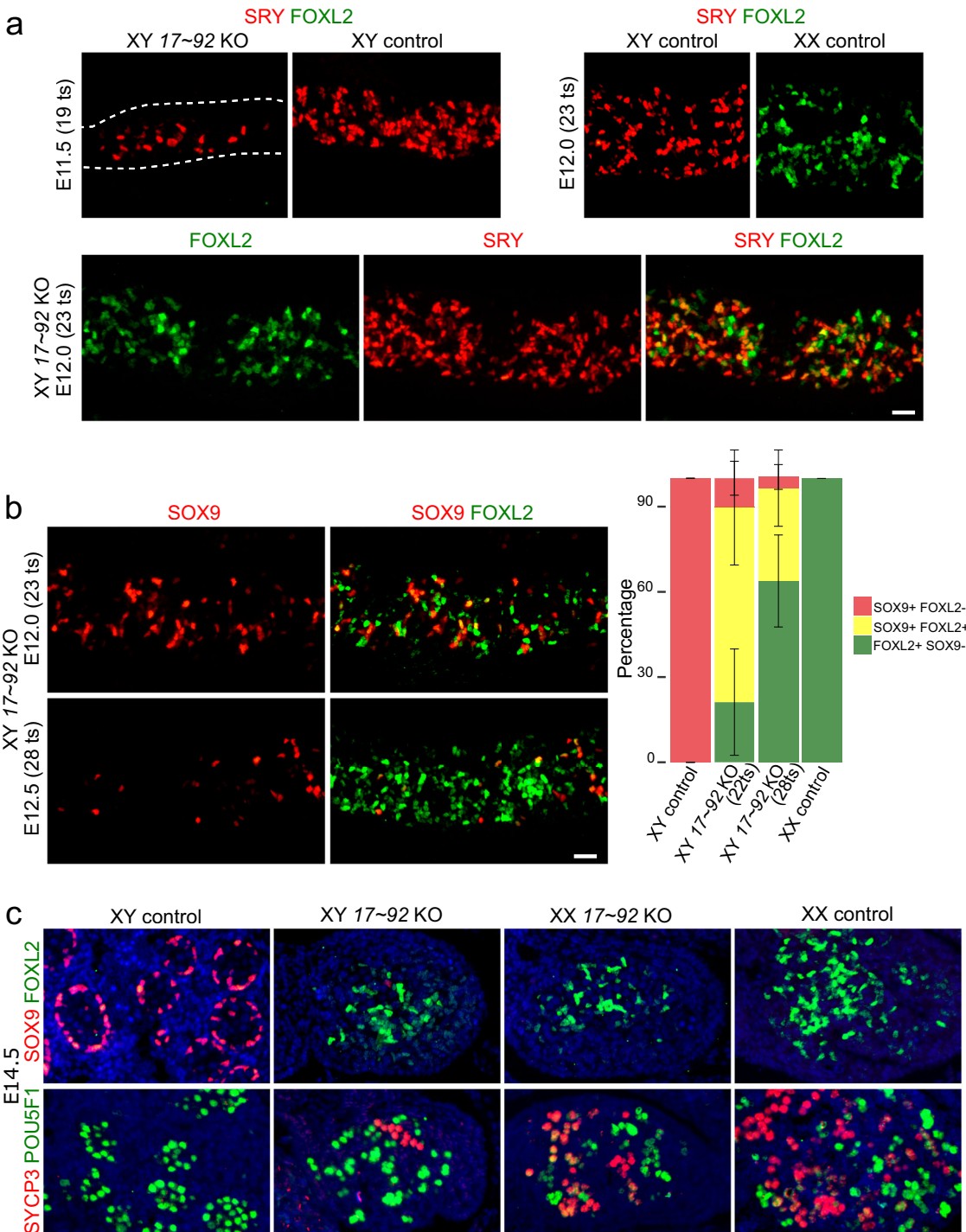

**Fig. 5 | Time course of granulosa and oocyte cell differentiation in XY *miR-17-92* KO gonads. a** Double immunofluorescence for SRY and FOXL2 in gonads during (E11.5) and shortly after the sex determination stage (E12.0). **b** Double immunofluorescence for SOX9 and FOXL2 after sex determination. Cells showing colocalization of both proteins are common at E12.0 but very scarce at E12.5. Data are presented as mean values ± SD. **c** Double immunofluorescences for SOX9 and FOXL2 (upper panel) and for SYCP3 and POU5F1 (lower panel) in gonads at E14.5.

SOX9⁺/FOXL2⁺ cells are almost completely absent at this stage, in which only FOXL2 expression persists. Meiotic oocytes (SYCP3⁺ cells) are seen in XY mutant gonads, evidencing complete ovarian differentiation. Similar results were obtained for the 3 animals of each genotype and developmental stage we analysed. The scale bars shown in the lower right images of panels (**a**), (**b**) and (**c**) represent 25 μm for (**a**) and (**b**), and 20 μm for (**c**), respectively. Mutant mice were generated using the Cre/LoxP system.

*miR-17-92* primes the bipotential gonadal primordium to ensure proper *Sry* expression timing and subsequent testicular differentiation. In summary, our study uncovers an unexpected role for the *miR-17-92* cluster and for miRNAs in controlling the early steps of mammalian sex determination.

## Methods

### Ethics
The research work presented here complies with all relevant ethical regulations and was first approved by the University of Granada Ethics Committee for Animal Experimentation and then by the regional

government, "Junta de Andalucía" under license number 16/12/2021/188, as well as by the Landesamt für Gesundheit und Soziales, Berlin, Germany under license number G0111/17. Research was performed in accordance with the relevant guidelines and regulations dictated by these committees.

## Transgenic mice

To generate mice (*Mus musculus*) with a null deletion of the *miR-17-92* cluster we crossed two months old mice with a floxed allele of *miR-17-92* on a C57BL/6 × 129S4/SvJae background acquired at the Jackson Laboratory (Bar Harbor, ME, USA; Stock 008458)[23] with *Tg(CAG-cre)*[INagy] mice (MGI:3586452[46]). The resulting *Tg(CAG-cre)*[INagy]*;miR-17-92*[+/del] offspring was backcrossed to *miR-17-92*[flox/flox] mice to obtain *Tg(CAG-cre)*[INagy]*;miR-17-92*[del/del] embryos. Figure 2a and Supplementary Fig. 2 show the effective deletion of the *miR-17-92* region in mutant mice. For timed pregnancies, plugs were checked every morning after mating and noon was taken as embryonic day (E) 0.5. Genotyping was carried out on genomic DNA derived from adult tails or embryonic yolk sacs using previously described primers[23,47]. We also generated mice carrying the homozygous deletion for *miR-17-92* using CRISPR/Cas9 combined with tetraploid complementation assays for embryo generation[24,25]. The single guides RNA (sgRNA) were designed with Benchling (https://www.benchling.com/crispr/) and cloned into the CRISPR/Cas9 pX459v2 plasmid (ref. 62988 Addgene). XY mouse embryonic stem cells (C57BL/6J-129 F1 hybrid background, G4 mESC) were cultured on a layer of CD1 fibroblasts and transfected with the sgRNA using FuGENE HD (Promega, Madison, WI, USA). Cells were split and transferred onto DR4 puro-resistant feeders after 24 h and selected for two days with puromycin. Then, clones were grown for 5–6 additional days, picked, and transferred into 96-well plates on CD1 feeders. After 2 days of culture, plates were split in triplicates (two for freezing and one for DNA harvesting). After genotyping, clones homozygous for the deletion were expanded and selected for tetraploid aggregation. Primers used for genotyping were as follows: 5´-GGACCTAGCAGCACCCGAAG-3´, 5´-GGGAGGAGCCAATAGCCAGA-3´ and 5´-GGAAAGATGGCAAACTGATGGT-3´, 5´-GGATCATGTGGATCGTGCTG-3´. The sgRNAs sequences were: 5´-CACCGTCGAGTATCTGACAATGTGG-3´ and 5´-CACCGGTCACTCACAATCAAACCC-3´.

To produce mice bearing a deletion of the *miR-17* seed sequence within the 3´ UTR of *Foxl2* (*Foxl2-3´-UTR*[del/del]) we used as well the CRISPR/Cas9 technology in combination with tetraploid aggregation assays as described above. In this case, primers used for genotyping were: 5´-GAAATCCCGTGACCTGGTGG−3´ and 5´- GCTAGCTGCTGAACTACGGT-3´. The sgRNA sequence was 5´-CACCGTTGTGTTTGTACGTGTGTG-3´.

Mice were housed under Specific Pathogen-Free (SPF) conditions in the animal facilities of the Center for Biomedical Research (University of Granada, Granada, Spain) and the Max-Delbrück Center for Molecular Medicine, Berlin, Germany. The animals had ad libitum access to food and water and were kept in groups. The occupancy density of the cages (microventilated) was in accordance with legal requirements. The cages were kept at a temperature of 22 +/− 2 °C, a humidity of 55 +/− 10% and a 12 h/12 h dark light cycle. The animals were provided with activity elements in the form of nest building material and hiding places.

## Histological and immunostaining methods

For histology, embryos were dissected, collected in PBS, fixed in Serra´s solution (ethanol: 37% formaldehyde: acetic acid, in proportions 6: 3: 1, respectively), embedded in paraffin, sectioned (5-μm thick), and stained with haematoxylin and eosin. For double immunofluorescence, the two primary antibodies (raised in two different mammalian species) were incubated overnight at 4 °C and then appropriated conjugated secondary antibodies were applied. Used antibodies are listed in Supplementary Data 22. Digital

photomicrographs were taken in a Nikon Eclipse Ti microscope (Nikon Corporation, Tokio, Japan).

## Morphometric analyses

The shape of the embryonic gonad is near cylindrical but has an irregular surface, including deep furrows, mainly in mutant embryos. Hence, its volume could not be accurately calculated using the formula of a cylinder. Instead, we measured the area of the gonadal image in a 10× photomicrograph taken from a medial (that showing the maximum gonadal area), longitudinal section. For this, we used the ImageJ application (https://imagej.nih.gov/ij/) set at a scale of 2940 pixels per mm (the value corresponding to images taken using a 10× objective). The contour of the gonad was encircled with a line using the *freehand selection* tool and its area (A) was measured in square mm (this operation was repeated twice for each gonad and the mean value was used for further calculations). Similarly, the maximum length (L) of the gonad was depicted with a line and measured in mm. The gonadal volume (V) was calculated using the formula: $V = (\pi A^2)/(4L)$. The volume of the two gonads was calculated in 4 control males, 2 control females, 3 mutant males, and 4 mutant females at the 18–19 ts stage. Means were compared using two-sided Student's *t* tests.

The mitotic index of the coelomic epithelium-derived cells was calculated by dividing the number of proliferating cells (Ki67+) by the total number of coelomic epithelium cells (WT1+) present in gonadal images photographed using a 20× objective. For this, double immunofluorescence preparations were performed on longitudinal sections of embryonic gonads from embryos both at the 14–16 ts stage (3 control males, 3 control females, 3 mutant males, and 3 mutant females) and at the 18–19 ts stage (4 control males, 2 control females, 3 mutant males, and 4 mutant females). Means were compared using two-sided Student's *t* tests.

## Analysis of apoptosis

Apoptotic cells present in gonadal histological preparations were revealed by the terminal deoxyribonucleotidyl transferase (TDT)-mediated dUTP-digoxigenin nick end labelling (TUNEL) assay using the *Fluorescent* In Situ *Cell Death Detection Kit* (Roche ref. 11684795910, Roche, Mannheim, Germany), according to the manufacturer's instructions. We omitted the enzyme solution for negative controls.

## Bulk RNA-seq

For RNA-seq, total RNA was purified from gonad-mesonephos complexes at E11.5 and from isolated gonads at E12.5 using the RNeasy Micro Kit (Qiagen, Hilden, Germany). For each genotype, 3 replicates were used, including 2 pair of gonads per replicate. Libraries were prepared with the NEBNext Ultra II Directional RNA Library Prep Kit for Illumina (New England Biolabs, Ipswich, MA, USA; E11.5 samples) and the TruSeq stranded mRNA Library prep (Illumina, San Diego, CA, USA; E12.5 samples) and sequenced with the Illumina Hiseq4000 platform (2×75 PE).

## Single cell RNA-seq

As crosses between *Tg(CAG-cre)*[INagy]*;miR-17-92*[+/del] and *miR-17-92*[flox/flox] mice only provide 1/8 XY KO mice, it would be very difficult for us to collect the gonad pools we needed for the sc-RNAseq experiments. Therefore, we opted to use tetraploid aggregation method, which produces litters of isogenic embryos from mESC. Three to four pairs of gonads and mesonephros per genotype, sex and stage were dissected out and pooled to perform the sc-RNA-seq experiments. XY CRISPR KO embryos were derived from G4 mESC modified via CRISPR/Cas, to delete the *miR-17-92* cluster. XY controls were derived from the same parental G4 line. XX controls gonads were obtained from crossings between C57BL/6J males and 129 females, to obtain a similar background as the G4 mESC used to generate XY wild type and the KO embryos (G4: C57BL/6J − 129 F1 hybrid background). The reason for

employing a breeding strategy is that, XX cells generally yield a low number of embryos in tetraploid complementation assays, and thus would not provide with enough material to perform scRNA-seq experiments. For XY *miR-17-92* CRISPR KO embryos we analysed two stages, E11.5 and E11.75, while for the XY and XX controls we only assessed the E11.5 embryonic stage. Individual cell suspensions were performed by using 0.05% Trypsin-EDTA (ref. 59417C, Sigma) for 7 min at 37 °C with pipetting every 2–3 min. Trypsin digestion was stopped by adding 5% BSA (ref. B9000S, New England Biolabs) and the cell suspension was passed through a 40 μm Flowmi cell strainer (ref. 15342931, Fisher scientific). Finally, cells were collected by centrifugation (300 × g for 5 min at 4 °C) and the pellet was resuspended in PBS for cell counting and viability check and then fixed with methanol (98%) added dropwise. Fixed cells were kept on ice for 15 min. and subsequently moved to −80 °C for extended preservation. After rehydration, about 10,000–20,000 cells were loaded on a 10× Chromium controller and sc-RNA libraries were prepared using the Chromium Next Gem Single Cell 3' Reagent Kit v3.1 (PN-1000128) following the manufacturer's instructions. Finally, libraries were sequenced using Illumina NovaSeq 6000 SP with the following parameters: paired-end, 28 + 8 + 91 cycles. The sequencing depth was 200 million reads per sample. Two technical replicates were performed per genotype except for the XY E11.75 KO mice, with one replicate.

### Bioinformatic analyses

The bulk RNA-seq reads were mapped to the mm10 mouse genome and counted with the "align" and "featureCounts" function from the R subread package[48]. Only genes with 1 or more cpm (counts per million) in at least two of the samples were considered to be expressed and used for further analysis. Analysis of differential gene expression was performed with edgeR[49] Genes were considered to be differentially expressed at a false discovery rate (FDR) < 0.05. Gene Ontology analysis was performed with the enrichGO function of the clusterProfiler bioconductor package[50]. General terms and redundant terms were not displayed. For gene-concept analysis we used the cnetplot function of the same suite. Prediction of target sites for the *miR-17~92* seed families was performed using TargetScan (mouse version 8)[51]. To analyse the gonadal cell lineage-specific expression of the individual members of the *miR-17~92* cluster we used the RMA normalised data from Jameson et al.[1] and differential expression was assessed with the limma R package. Regarding single cell RNA-seq, each single cell library was aligned to the mm10 reference genome (mm10_v3.0.0, obtained from 10× Genomics) and transcriptome (modified to include a second exon of *Sry* as stated in previously[26] and filtered with Cell Ranger Software; version 6.1.1) using default parameters. The filtered counts matrices of all the samples with two technical replicates were merged using Seurat (version 4.0.2)[52] before proceeding with downstream analyses. For the male samples, once they were merged, thresholds for percentage of mitochondrial genes were set in order to filter low quality cells (Male control 11.5 = 10%, Male mutant 11.5 = 10%, Male mutant 11.75 = 9%). Also, blood cells were filtered from the analyses according to their percentage of beta globin transcripts (male control 11.5 = 1.43%, male mutant 11.5 = 0.39%, male mutant 11.75 = 0.73%). Finally, only cells with nCount_RNA < 30000 (transcripts) and nFeatures_RNA < 7000 (genes expressed) were considered. After quality check we analysed 23527, 17057, 8492, and 3669 cells from gonads of XY controls at E11.5, XY *miR-17-92* mutants at E11.75, XY *miR-17-92* mutants at E11.5 and XX controls at E11.5, respectively. Next, data was normalised using default settings, 2000 variable features were identified using the default Seurat method (vst). Data was then scaled regressing the nCount_RNA (transcripts) and percentage of mitochondrial genes. Finally, clusters were obtained with a resolution of 1.2 with the FindClusters function from Seurat. For the female samples, the filtering was made at this step by excluding those clusters with low quality (low nFeatures_RNA or high number of beta haemoglobin

genes expressed, compared to the rest of the clusters). Doublets were excluded using DoubletFinder (version 2.0.3)[53]. In order to generate a single dataset with all samples, the Seurat objects were integrated using the Canonical Correlation analysis (CCA) method[54] with 40 dimensions and using the 2000 most variable features. Then, the integrated object was scaled as previously stated and clusters were called at resolution 1 with the FindClusters function from Seurat. A single cluster was manually removed (it contained cells only from control males at 11.5 and was scattered along all the UMAP). Markers for each cluster were identified using the FindMarkers function of Seurat with default parameters. Clusters were annotated according to their expression markers according to previous bibliography[26]. Finally, DEG lists between the clusters and samples of interest were generated using the Model-based Analysis of Single cell Transcriptomics (MAST) method[55]. Mitochondrial and ribosomal genes as well as blood markers were manually removed from the DEG lists.

### Reporting summary

Further information on research design is available in the Nature Portfolio Reporting Summary linked to this article.

## Data availability

The data supporting the findings of this study are available from the corresponding authors upon request. Transcriptomic data are publicly available on Gene Expression Omnibus under the following accession numbers: Serie: GSE225677 [https://www.ncbi.nlm.nih.gov/geo/query/acc.cgi?acc=GSE225676]; Subserie RNA-seq: GSE225675; Subserie scRNA-seq: GSE225676. Source data for the figures and Supplementary Figs. are provided as a Source Data file. Source data are provided with this paper.

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

## Acknowledgements

We are indebted to the Animal Experimentation Unit of the University of Granada´s Centre for Scientific Instrumentation for kindly extending the maintenance of our mouse colony temporally after the end of our financial support for this work. We thank the sequencing core, transgenic unit and animal facilities of the Max Delbrück Centre for Molecular Medicine for technical assistance. We thank R. Kühn, C. Scholl, M. Altmann, G. Kussagk, S. Bomberg, S. Reissert-Oppermann and C. Westphal for their support with the transgenic work, and Dr. Dagmar Wilhelm for kindly providing us with the SRY antibody. This work was supported by grants from the Andalusian Government (Junta de Andalucía), P20_00583 to R.J. and F.J.B. and P11-CVI-7291 to M.B., and by grant no. PID2022-139302NB-I00 from the Spanish "Agencia Estatal de Investigación" to F.J.B. and R.J. Research in the Lupiañez lab was funded by the Deutsche Forschungsgemeinschaft (International Research Training Group 2403, including PhD fellowship to I.M-G), by the European Research Council (grant no. 101045439, 3D-REVOLUTION) and by the Spanish "Agencia Estatal de Investigación" (grant no. PID2022-143253NB-I00/ AEI/10.13039/501100011033/ FEDER, UE). The authors thank the Spanish Ministry of Science and Innovation for the PhD fellowships granted to A.H. and M.L.

## Author contributions

R.J. and F.J.B. conceived the project; R.J., F.J.B., D.G.L., and M.B. designed the experiments; R.J., F.J.B., A.H., I.M.-G., M.L., F.M.R., and J.J. performed laboratory experiments; F.J.B., M.B., and I.M.-G. carried out bioinformatic analyses; R.J., F.J.B., M.B., and D.G.L. collected the financial support; R.J. and F.J.B. wrote the first draft of the manuscript and all authors contributed to the final version.

## Funding

## Competing interests

The authors declare no competing interests.
