## [Peer Review File · Nature Communications]

Complete male-to-female sex reversal in XY mice lacking the miR-17~92 clusterREVIEWER COMMENTS

Reviewer #1 (Remarks to the Author):

In this manuscript, Hurtado, Mota-Gomez, and Real describe a striking phenotype caused by homozygous deletion of miR-17~92 in mice. They report that in the absence of this miRNA cluster, XY embryos fail to develop male gonads and instead develop female gonads (sex reversal). By performing bulk and single cell RNA sequence, the authors show that this phenotype is associated and probably caused by delayed expression of SRY, which in turn results in delayed and reduced expression of one of its key targets: Sox9.

The phenotype is clear, and the molecular and histological characterization is compelling. The lack of a detailed molecular mechanism through which miR-17~92 temporally controls SRY expression in the developing gonads somewhat reduces the impact of this work, but this is a common theme in the miRNA field and reflects the complexity and subtlety through which miRNA modulate the expression of complex gene networks and is certainly not a fault of the authors. Indeed, the authors efforts to mutate the miR-17 binding site in Sox9 to test a potential linear pathway linking miR-17~92 to sex determination must be commended, despite the negative result.

The topic is certainly of substantial interest to the broad readership of Nature Communication, the relevant scientific literature is cited, and the computational analysis performed by the authors is exhaustive and well carried out. The figures are clear and the results are consistent with the model proposed by the authors.

Since the authors have the miR-17~92 floxed animals, conditional ablation of the cluster in the developing gonads would have been a nice addition to this manuscript, as it would have bypassed the perinatal lethality caused by miR-17~92 deletion. But I would not insist on including this time consuming experiment.

I have a couple of comments that I would however like to see addressed before publication:

- 1) The rationale for switching from the well characterized miR-17~92 KO allele used in Figure 1 to the CRISPR-KO allele generated in ES is not spelled out in the manuscript. The generation and characterization of this new allele is also not described in the manuscript (and I couldn't find details in the material and methods section). I feel this is something that the authors must address in a revision.
- 2) I assume the authors used XY ES cells to generate this allele, so tetraploid complementation cannot give female embryos. What was the origin of the XX gonads used in the single cell RNA seq experiments described in the paper? Why not use XX and XY littermates from the germline KO allele?
- 3) Given the number of different clusters in the UMAP figures, using just colors to identify them can be challenging for many people (including this reviewer!). I would recommend that the authors identify the clusters also by number.

Reviewer #2 (Remarks to the Author):

In this work Hurtado, Mota-Gómez, Lao and colleagues use genetically engineered mouse models to study the impact of miRNAs in sex determination in mammals. They show that loss-of-function for the highly-conserved miR-17~92 cluster results in sex-reversal of male to female in mouse embryos. This is a very interesting and intriguing finding given that this cluster has been extensively studied in mice due to its wide implications in human diseases but such phenotype had not previously been described. As such, I believe that this work would be of general interest to the research community.

I do however have a few comments and suggestions that follow below. Note that my expertise lies in miRNA biology and mouse genetics. As such, I am not able to judge the work related to gonadal development, and hope those expertise will be provided by an independent reviewer.

Overall, it is my opinion that the manuscript would benefit from a re-structuring that better places the current work in the context of previous literature, guides readers through normal gonadal development and how it is disrupted in the mutants, and places less emphasis on trying to identify direct targets of miR-17~92, and more emphasis in clarifying and supporting their hypothesis that compromised proliferative pathways underly the phenotypes described here. Controls are lacking in same places and need to be added to strengthen the authors approaches and conclusions.

1. Line 34 (abstract): the authors write "Consistent with known mechanisms of miRNA-mediated gene regulation, the expression of miR-17~92 target genes is not stabilized in undifferentiated XY mutant gonads" this should be changed to "Consistent with known mechanisms of miRNA-mediated gene regulation, the expression of miR-17~92 predicted target genes is upregulated in undifferentiated XY mutant gonads" since the authors provide no measurements of RNA stability in this work and have not validated most predicted targets.

2. line 75 (intro): the authors write "As a consequence of this heterochrony, pre-supporting cells undergo a transient state of sexual ambiguity that results in failed differentiation into Sertoli cells and the subsequent activation of the ovarian program". The authors do not show a causal link between deregulation of Sry expression timing in miR-17~92 mutants and the sex-reversal phenotypes. Because of that, the authors should rephrase this sentence to indicate that this is a hypothesis.

3. The motivation behind the work could be made clearer to help readers follow the work and how it relates to the current status of the field. In the discussion the authors write (line 319): "Since miRNAs discovery, numerous attempts have been made to elucidate their role during sex determination. However, to date, none of them could demonstrate that such post-transcriptional regulators can play a fundamental role in this process^{9,6}. Indeed, previous studies compromising the biogenesis of these molecules suggested that they are dispensable for sex determination^{15,16}." However, later in the discussion they write (line 342) "In fact, the miRNAs of the miR-17~92 cluster are evolutionarily well conserved, appearing around 500 Mya in primitive vertebrates, where they have been shown to regulate genes involved in gonadal development and sex change⁴³." So which one is it? Have miRNAs been or not implicated in sex-determination? And is the work cited in reference 43 the motivation for the current work? What model system was used there? And what were the findings? How do they differ from the current work? Answers to these questions should be provided, and likely added to the introduction to help place the current work in context.

4. Given that this work is likely aimed at a general audience, most of which will not be familiar with gonadal development in mice, the authors should place a bigger emphasis in educating their readers in the main aspects of this process either in the introduction or in the first section of the results. Particularly, a schematic representation of gonadal development over the stages addressed in this manuscript clarifying both the cell types that are present, their markers, and their relationships would be important for readers to be able to follow the work. What are Sertoli cells? Or granulosa cells? Or the "supporting cells"? What is the relationship between Supporting cells and Sertoli cells? Why would reduction of one affect the other? It would also be helpful to label some of the the structures/cell types in the IFs as well.

5. line 90: "Histological analyses on XY KO gonads revealed a complete absence of male-specific structures, such as testicular cords." This data should be shown. I assume by histology the authors are referring to tissue sections which are not included in the manuscript.

6. The authors claim that miR-17~92 KO males display complete sex reversal (line 97), with early

signs of ovarian development and germ cell meiosis (line 95). Do these animals show oocyte maturation? If so, can you show examples? If not, can this be called complete sex reversal?

7. Figure 2a: instead of log₂-fold changes can you show an unsupervised clustering of the samples based on the genes with dimorphic expression? This will give a better sense of how the KO samples segregate relative to controls.

8. Line 128: "Gene Ontology analysis identified several common GO terms between E11.5 mutant and controls, in both XX and XY backgrounds (Extended Data Fig. 2b; Extended Data Tables 10 and 11)". This sentence is confusing and should be re-written. The GO terms are not common between mutants and controls. They are enriched in when looking at genes deregulated between WT and KO animals in both XX and XY backgrounds.

Line 132: "In particular, we found GO terms related to cell proliferation and molecular pathways involved in its control." Please specify what terms you refer to in the text.

9. The authors find that the genital ridge of the mutant gonads contains less proliferative progenitor cells at E11.0. What is the genital ridge and how does it relate to the lower abundance of Sertoli cells at 11.5?

10. Line 162: I am very confused on why after characterizing the sex-reversal phenotype in KO animals generated through recombination of a floxed mouse strain the authors have chosen to follow up their studies with embryos derived from tetraploid aggregation following deletion of the cluster in mouse ESC. This strategy allows researchers to generate "fully ESC-derived embryos" by complementing the ESC with tetraploid embryos that can only give rise to extra-embryonic tissues. While a useful strategy to study embryonic development when mutations in the extra-embryonic tissues lead to embryonic lethality, they provide no added benefit to the current study. If anything, they can contribute to additional confounders in the data as the embryonic development will be highly dependent on the ESC quality. The authors should clarify why tetraploid complementation was used for these studies. In addition, they need to provide a full characterization of the ESCs and how the miR-17~92 modification was accomplished. From Figure 2 it seems the authors modified both males and female ESCs. Is the deletion in both lines identical? How many clones per targeting were tested? What are the authors using as controls for these experiments? Embryos generated by tetraploid complementation using parental ESC lines? Or Embryos generated by tetraploid complementation using ESC lines that underwent a "mock" targeting? The authors say "these mutants displayed an identical phenotype to those previously generated via Cre-Lox breeding" but this data also needs to be provided.

11. Line 168: please clarify in the main text (not just the figure) which embryonic stage was profiled by single-cell sequencing. Also, what tissues were used for this experiment? I assume dissected gonads? But it is not described anywhere.

12. The authors note that Sertoli cells are highly reduced in KO male gonads. Do the authors think this is a consequence of the proliferation defects observed earlier in the genital ridge? The authors say the lack of Sertoli cells indicate a defect in the timing of their differentiation. What are the precursors for Sertoli cells? Do they display defects in proliferation?

From the discussion it seems the authors favor a deregulation of the MAPK pathway as a mechanism driving sex reversal. This would certainly fit with roles of the miR-17~92 cluster in other tissues and cell lines and it seems also with sex reversal literature. The manuscript would benefit from exploring this aspect further. Particularly, it would be important to show not only de-regulation of genes of this pathway but also that the pathway itself is deregulated in mutants (by IF or western), and to define both the timing of that deregulation and how it relates to the morphological (i.e. lack of Sertoli cells) and molecular (i.e. timing of Sry onset) defects described here, as well as which cell types are

affected.

In the discussion (line 330) the authors write: "This proliferative reduction in mutant gonads appears to affect all progenitor cell types similarly, as their relative numbers do not differ significantly from those of controls, notably in the case of the pre-supporting cell lineage (Fig. 2f)." Can the authors confirm that that is the case (perhaps by co-IF with progenitor cell markers or by FACS)? "Hence, the number of pre-Sertoli cells present at the time of sex determination (E11.5), although reduced, should be sufficient to initiate testis differentiation." Why? Is it known what is the minimum number of pre-sertoli cells required to support testis differentiation? Are there markers for pre-sertoli cells? If so, can authors quantify and characterized them? Are pre-sertoli cells the same as pre-supporting cells? The authors use both terms to refer to progenitors of Sertoli cells.

What is the relationship between Sertoli cells and Sry expression? The authors write (line 336): "The lack of Sertoli cells in XY miR-17~92 mutant gonads at the sex determination stage is associated with the heterochronic expression of Sry, which must act within a critical time window to activate Sox9 and trigger the male pathway³⁶". Do Sertoli cells express Sry? Or is Sry expression required for Sertoli cell differentiation? Please clarify.

Aside from expanding and clarifying this section of the work, I would also re-structure the manuscript to make this hypothesis clearer as it seems the most likely explanation for the defects described here.

13. The authors consistently describe de-regulation of miR-17~92 targets in the manuscript. These should be modified to "predicted" or "putative" targets whenever there is no direct functional validation of targeting.

I would suggest the authors place less emphasis on trying to understand what the direct targets of miR-17~92 driving this phenotype are. It is now well established that miRNAs affect hundreds of genes modestly and that in most cases no single target de-regulation will be able to recapitulate the phenotypes of a miRNA KO. It is highly commendable that the authors attempted to validate Foxl2 as the direct target in this case. However, I would suggest the authors give less emphasis to this section (given that it yields a negative result), moving it down or potentially to a supplementary note. As is, lines 294-307 of the manuscript disrupt the section describing the FOXL2 SOX9 double positive cells. If keeping this section, the same question related to the ESC targeting and controls that I raised regarding the previous tetraploid aggregation experiments need to be included. In addition, the authors should clarify what type of binding site this is (8mer? 7mer-m8? 7mer-A1?). In addition to explaining the source of the control embryos for these experiments, the authors need to include controls of the same developmental ages as the binding-site mutants. For example, Sup Figure 7b shows sequencing data from mutants but not controls.

14. Controls are also missing in Figure 4a (E11.5 stage).

Minor comments:

- Extended Data Figure 1: Mir17hg refers to the human gene. This should be corrected as the figure describes the mouse gene.
- Extended Data Figure 1: the authors show the sequences of the mature miRNAs, but label them as "target sequence". This may lead readers to think it refers to the sequence of the miRNA target. Label should be changed to "miRNA sequence" to make things clearer.
- Extended Data Figure 1: (b) label is missing. In addition, it is unclear what measurements are being plotted in the graphs. What does log intensity value refer to? The figure legend says this data comes from the work of Jamerson and colleagues, but readers should not have to go to that publication to understand the data shown here. Please clarify. In addition, in page 2 lines 68-69 ("Interestingly, miR-17~92 cluster members are expressed in mouse gonads from both sexes during and after the sex

determination stage (Extended Data Fig. 1b)”) the authors should add a reference to the Jamerson work when referring to this data.

- Line 227: process instead of “process”
- Line 242: should maintaining be replaced by “repressing”?

Reviewer #3 (Remarks to the Author):

In mice, expression of the testis-determining gene Sry must reach a threshold for a short period of time to induce differentiation of the testis from the bipotential gonad. The regulatory mechanisms regulating Sry expression have been extensively studied showing that the activation of Sry involves the phosphorylation of GATA4 by the mitogen activated protein kinase (MAPK) pathway. A delay in Sry expression or an expression level that is too low, results in sex reversal. The authors show the potential involvement of a new factor, the miR17-92 in the regulation of Sry in mice. The delay of Sry expression is well documented in the miR17-92 mutants but the causes remain elusive.

This is a potentially very interesting finding. However, it remains to be demonstrated that the phenotype does not result from a global effect of growth retardation but from a specific gonad defect. If it is not such a defect that leads to the sex reversal phenotype, the authors need to explain why there is a delay in Sry activation.

Major issues:

The results are based on conditional deletion of miR17-92, and although the sex reversal phenotype is clear, the level of miR17-92 deletion remains to be demonstrated by quantitative PCR experiments. The authors need to show the efficacy of the recombination of the CAG-CRE on miR17-92 (Figure 1). This is a very important point because the entire study and conclusions are based on the impact of miR17-92 deletion.

Why did the authors use two different models, an inducible deletion model and a tetraploid model? Would crosses between mutants heterozygous for the knock-out allele not have produced homozygotes? Are homozygous embryos lethal very early and what is the phenotype? This is a source of confusion, and it would be advisable to clarify in each experiment which model was used and why?

The embryos die before birth suggesting a pleiotropic effect of conditional deletion of miR17-92 in the embryo or in the placenta. In Figure 1, the kidneys are smaller (this is particularly obvious in the XX mutants). It is important to discriminate whether the reduction in miR17-92 level affects the whole embryo or whether it affects the gonad more specifically. Indeed, a developmental delay could indirectly impact Sry expression.

The authors need either to use a cre recombinase that specifically delete miR17-92 in gonads, or they need to study the development of other organs.

If the authors find that there is no growth retardation of the embryos, what is the explanation for the delay of Sry expression? The main regulators of Sry is the MAPK pathway including activation by Gadd45g and phosphorylation of GATA4. Gadd45d shows reduced expression in miR17-92 mutant gonads. The labs of A.Greenfield and C.Niehrs have studied the impact of Gadd45g deletion on phosphorylation of p38MAPK and GATA4 thus explaining the delay of Sry expression (Dev Cell 2012 and after). Does the delay in Sry expression result from reduced GATA4 phosphorylation and reduced activation of the MAPK pathway?

Another important step in testis development is the epigenetic status. The authors show that abundant double positive cells for SOX9 and FOXL2 are detected in the mutant gonads. Similar observations have been reported in the Cbx2^{-/-}, a member of the PRC1 complex involved in epigenetic regulation.

The regulation of gonad differentiation by miR17-92 is potentially a very interesting discovery, but it remains to be clarified how this miR is involved in sex determination (the MAPK pathway, histone methylation, both, another mechanism?).

Additional comments

Line 24 whereas its absence in XX embryos leads to ovarian differentiation.

Authors need to be more cautious in describing previous results. Some of the results obtained in mice may not be true in all mammals, and in humans. In humans, mutations in the SRY gene results in 46,XY complete gonadal dysgenesis, which is not ovarian differentiation. I suggest that the authors replace mammals (line 22) by mice and SRY by Sry or ovarian differentiation by complete gonadal dysgenesis.

Same comment lines 40 to 44.

Line45. Reference 8 is not relevant. The downregulation of WNT signaling in XY gonads has been clarified and involves ZNRF3 as evidenced by Harris/Greenfield (2018).

Line 50 remains instead of remain

Line 59. Huang and Yao have used the Sf1-cre (Bingham et al, 2006) to delete Dicer 1.

Manuylov/Tevosian (2011) have shown that the Sf1Cre recombined inefficiently in coelomic epithelial cells and is not very efficient around E11.5. This might explain the absence of an obvious phenotype when Dicer is deleted with this cre recombinase.

Line 84 Although the cre recombinase used by the authors is described in Mat and Met, it should be added in the results section. The gene excision data is confusing (The legend of extended Data Figure 10 should be more detailed). The authors' current attempts to clarify this point have to be better shown. This is important because excision efficiency is key to the authors' findings. The authors need to quantify the level of miR17-92 deletion using quantitative RT-PCR as previously described in Hurtado/Barrionuevo (2018).

Can the authors comment why did they not use the mating between miR17-92^{+/-} heterozygous mice to get homozygous miR17-92^{-/-} embryos?

Figure 1:

The development of the embryos seems to be impaired by genetic deletion of miR17-92. The impact of miR17-92 deletion in the whole embryos/other tissues of the embryos need to be documented (see my comments above)

The official name of OCT4 is POU5F1.

Line 97

XY mice undergo complete primary sex reversal.

Line 110 Contrarily, all germ cells of the other three genotypes had entered meiosis. How did the authors show that all germ cells have entered meiosis?

Figure 2

Figure2c. WT1 is expressed in the gonad and the mesonephros (Liu/Yao 2016, Sasaki/Saitou 2021, Mayere/Nef 2022). In these images, WT1 immunostaining is unexpected because it seems to be

restricted to the coelomic epithelium in XY control at E11.0 and not expressed in the mesonephros at E11.0 and after. Is there an effect of genetic deletion of miR17-92 on WT1? This needs to be clarified.

Figure 2 d. For clustering and identification of populations, cells were coloured by genotypes (d). The authors show only a merged image, which is difficult to evaluate. The authors should show images for each color/genotype separately together with merged images so that the readers can examine each genotype appropriately.

Figure 2f. The pre-granulosa cells are not represented whereas immunostaining of FOXL2-positive cells show that these cells are present around E11.75 as shown in Figure 4b. Are they present or absent in the single cell RNA seq analysis? If they are absent, can the authors comment on the discrepancy between Figure 2 and 4?

All along the text, the authors mainly used the term of "granulosa cell" to design the female supporting cell lineage. Are they sure this term is appropriate before follicle formation occurs? Pre-granulosa cell seems commonly used before follicle formation.

Figure 3: the delay of SRY is convincing. It doesn't have to be backed up by statistics. For small sample sizes (n=3) descriptive statistics can be not appropriate. Please check the policy of the journal.

Figure 4: While the expression of FOXL2 and SRY can be observed when SRY is delayed and in turn SOX9, the double positive cells FOXL2/SOX9 are usually rare. (Garcia-Moreno/Capel 2019) showed that cells in the *Cbx2*^{-/-} XY gonad express SOX9 and FOXL2 simultaneously, with a reduction in SOX9-positive cells over time. Their results suggest that the initiation of the testis program occurs, but the stabilization phase, in which the female pathway is silenced by epigenetic chromatin regulation, is disrupted in the absence of CBX2. It is thus conceivable that the phenotype due to reduced miR17-92 expression could modify the epigenetic state of chromatin. Is it a cause or a consequence?

Figure 4b. How were the cell percentages quantified, and how many cells were counted?

REVIEWER COMMENTS

Reviewer 1 (Remarks to the Author):

In this manuscript, Hurtado, Mota-Gomez, and Real describe a striking phenotype caused by homozygous deletion of miR-17~92 in mice. They report that in the absence of this miRNA cluster, XY embryos fail to develop male gonads and instead develop female gonads (sex reversal). By performing bulk and single cell RNA sequence, the authors show that this phenotype is associated and probably caused by delayed expression of SRY, which in turn results in delayed and reduced expression of one of its key targets: Sox9.

The phenotype is clear, and the molecular and histological characterization is compelling. The lack of a detailed molecular mechanism through which miR-17~92 temporally controls SRY expression in the developing gonads somewhat reduces the impact of this work, but this is a common theme in the miRNA field and reflects the complexity and subtlety through which miRNA modulate the expression of complex gene networks and is certainly not a fault of the authors. Indeed, the authors efforts to mutate the miR-17 binding site in Sox9 to test a potential linear pathway linking miR-17~92 to sex determination must be commended, despite the negative result.

The topic is certainly of substantial interest to the broad readership of Nature Communication, the relevant scientific literature is cited, and the computational analysis performed by the authors is exhaustive and well carried out. The figures are clear and the results are consistent with the model proposed by the authors.

Since the authors have the miR-17~92 floxed animals, conditional ablation of the cluster in the developing gonads would have been a nice addition to this manuscript, as it would have bypassed the perinatal lethality caused by miR-17~92 deletion. But I would not insist on including this time consuming experiment.

We thank the reviewer for the positive comments and suggestions for improvement.

I have a couple of comments that I would however like to see addressed before publication:

1) The rationale for switching from the well characterized miR-17~92 KO allele used in Figure 1 to the CRISPR-KO allele generated in ES is not spelled out in the manuscript. The generation and characterization of this new allele is also not described in the manuscript (and I couldn't find details in the material and methods section). I feel this is something that the authors must address in a revision.

We apologize for the lack of clarity in this matter. Most of the experiments presented in this manuscript were performed through breeding between miR-17~92flox/flox and TG-Cag-Cre mice, to obtain homozygous specimens that lack the miR-17~92 cluster. However, for scRNA-seq experiments, we used an alternative strategy by generating homozygous KO embryos through CRISPR and tetraploid aggregation methods. The reason for using this strategy is that each scRNA-seq experiments would require the pooling of gonads from several mutants and, unfortunately, the crosses using Cre-LoxP breeding would only produce on average 1 out of 8 embryos with the appropriate genotype (XY homozygous mutants). Such reduced yield results in 0-1 mutants per litter, making highly unfeasible to collect several mutant gonads at once. In contrast, the tetraploid

aggregation method produces litters of isogenic embryos. Therefore, all the resulting embryos have the genotype of the modified ES cells (XY homozygous *miR-17~92* KO) generated through CRISPR. In our experimental design, we made a particular emphasis in that the sgRNAs were located at a equivalent position as the LoxP sites from the flox mice, such that the resulting deleted regions has analogous coordinates. Moreover, the mutants generated by CRISPR/Cas displayed the same phenotype as those obtained with the Cre-LoxP strategy. In our opinion, rather than being a problem, this has an added value as we demonstrate that the phenotype is consistent even if different approaches are employed to generate the mutants.

We have now included the corresponding explanation in the main text and in the material and methods section (lines 196-198, 202-203, and 516-530 of the revised version of the manuscript). We have also composed a new supplementary figure to showcase the phenotype of the CRISPR mutants (Extended Data Fig. 3 in the revised version of the manuscript).

2) I assume the authors used XY ES cells to generate this allele, so tetraploid complementation cannot give female embryos. What was the origin of the XX gonads used in the single cell RNA seq experiments described in the paper? Why not use XX and XY littermates from the germline KO allele?

We used the G4 mESC line (C57BL/6J - 129 F1 hybrid background, kindly provided by Andras Nagy lab), which has an XY genotype. Therefore, this cell line only produces XY embryos, and it was used to generate the data from XY KO mutants and wildtype controls. The origin of the XX gonads used in the scRNA-seq experiment came from crosses between C57BL/6J males and 129 females. Therefore, the embryos generated through these crosses have the same background as the G4 mESC used to generate XY wild type embryos and the KO via CRISPR/Cas9. An alternative would have been to use XX mESC with a C57BL/6J - 129 F1 hybrid background. However, XX cells generally yield a low number of embryos in tetraploid complementation assays, and thus would not provide us with enough material to perform this experiment. Even though XX gonads are derived from breeding, this sample integrates perfectly, and as expected, with the XY gonadal data.

We have included this information in the material and methods section (lines 516-530 of the revised version of the manuscript).

3) Given the number of different clusters in the UMAP figures, using just colors to identify them can be challenging for many people (including this reviewer!). I would recommend that the authors identify the clusters also by number.

We have modified Figure 2d and Ext. Data Fig 3a (Ext. Data Fig 4a in the revised version of the manuscript) accordingly.

Reviewer 2 (Remarks to the Author):

In this work Hurtado, Mota-Gómez, Lao and colleagues use genetically engineered mouse models to study the impact of miRNAs in sex determination in mammals. They show that loss-of-function for the highly-conserved miR-17~92 cluster results in sex-reversal of male to female in mouse embryos. This is a very interesting and intriguing finding given that this cluster has been extensively studied in mice due to its wide implications in human diseases but such phenotype had not previously been described. As such, I believe that this work would be of general interest to the research community.

We are grateful to the reviewer for the positive comments and the suggestions for improvement.

I do however have a few comments and suggestions that follow below. Note that my expertise lies in miRNA biology and mouse genetics. As such, I am not able to judge the work related to gonadal development, and hope those expertise will be provided by an independent reviewer.

Overall, it is my opinion that the manuscript would benefit from a re-structuring that better places the current work in the context of previous literature, guides readers through normal gonadal development and how it is disrupted in the mutants, and places less emphasis on trying to identify direct targets of miR-17~92, and more emphasis in clarifying and supporting their hypothesis that compromised proliferative pathways underly the phenotypes described here. Controls are lacking in same places and need to be added to strengthen the authors approaches and conclusions.

We apologize for the lack of clarity in certain sections. Following the reviewer's recommendation, we have made an effort to adapt the manuscript to a broader audience. Therefore, we have expanded the introduction to provide more background information on the process of sex determination in mammals (lines 43-64 of the revised version of the manuscript).

Regarding the reviewer's comment to put "*less emphasis on trying to identify direct targets of miR-17~92*", we feel that this is an important aspect of our study. It is quite uncommon that the disruption of a gene, specially a cluster of miRNAs, leads to complete sex reversal. Therefore, it is relevant to explore how *miR-17~92* may influence the expression of key genes involved in sex determination (including the *Sry* gene). We believe that this information will be of interest not only for experts in sex determination (like reviewer 3), but also in other fields like gene regulation by miRNAs.

1) Line 34 (abstract): the authors write "*Consistent with known mechanisms of miRNA-mediated gene regulation, the expression of miR-17~92 target genes is not stabilized in undifferentiated XY mutant gonads*" this should be changed to "*Consistent with known mechanisms of miRNA-mediated gene regulation, the expression of miR-17~92 predicted target genes is upregulated in undifferentiated XY mutant gonads*" since the authors provide no measurements of RNA stability in this work and have not validated most predicted targets.

We thank the reviewer for pointing out this mistake. We have changed the sentence accordingly (lines 33-37 of the revised version of the manuscript).

2) line 75 (intro): the authors write "*As a consequence of this heterochrony, pre-supporting cells undergo a transient state of sexual ambiguity that results in failed differentiation into Sertoli cells and the subsequent activation of the ovarian program*". The authors do not show a causal link between deregulation of *Sry* expression timing in *miR-17~92* mutants and the sex-reversal phenotypes. Because of that, the authors should rephrase this sentence to indicate that this is a

hypothesis.

Previous research has demonstrated that a delayed expression of *Sry* can result in sex reversal (see Hiramatsu et al., 2009; <https://doi.org/10.1242/dev.029587>). Based on this evidence, although we have not proven a causal link in our mutants, we believe that this is likely the underlying mechanism of sex reversal.

Therefore, we have rephrased the sentence as follows: “*As a consequence of this heterochrony, pre-supporting cells undergo a transient state of sexual ambiguity that likely causes failed differentiation into Sertoli cells and the subsequent activation of the ovarian program*” (Line 101 of the revised version of the manuscript). We hope that this is an acceptable statement that frames better our results in the light of previous knowledge.

3. *The motivation behind the work could be made clearer to help readers follow the work and how it relates to the current status of the field. In the discussion the authors write (line 319): “Since miRNAs discovery, numerous attempts have been made to elucidate their role during sex determination. However, to date, none of them could demonstrate that such post-transcriptional regulators can play a fundamental role in this process^{9,6}. Indeed, previous studies compromising the biogenesis of these molecules suggested that they are dispensable for sex determination^{15,16}.” However, later in the discussion they write (line 342) “In fact, the miRNAs of the miR-17~92 cluster are evolutionarily well conserved, appearing around 500 Mya in primitive vertebrates, where they have been shown to regulate genes involved in gonadal development and sex change⁴³.” So which one is it? Have miRNAs been or not implicated in sex-determination? And is the work cited in reference 43 the motivation for the current work? What model system was used there? And what were the findings? How do they differ from the current work? Answers to these questions should be provided, and likely added to the introduction to help place the current work in context.*

We apologize for creating such confusion. The first cited reference refers exclusively to mammals, whereas the second refers to non-mammalian vertebrates, so there is no contradiction here.

To better highlight this difference, we have modified the first sentence to make clear that we are referring to mammalian sex determination: “*Since miRNAs discovery, numerous attempts have been made to elucidate their role during mammalian sex determination. However, to date, none of them could demonstrate that such post-transcriptional regulators can play a fundamental role in this process. Indeed, previous studies compromising the biogenesis of these molecules in mouse gonads suggested that they are dispensable for sex determination*” (lines 373 and 375 of the revised version of the manuscript).

Reference 43 is a review on the evolution of *miR-17~92* and the frequent connections that this cluster has with processes related to sex development and sex reversal in non-mammalian vertebrates, which is quite relevant from an evolutionary point of view. This, together with the facts that these miRNAs are expressed in mouse embryonic gonads and that they have important roles in mouse adult testis, motivated us to perform this study. A new sentence in this regard has been included in the introduction section (lines 91-93 of the revised version of the manuscript).

4. *Given that this work is likely aimed at a general audience, most of which will not be familiar with gonadal development in mice, the authors should place a bigger emphasis in educating their readers in the main aspects of this process either in the introduction or in the first section of the results. Particularly, a schematic representation of gonadal development over the stages addressed in this manuscript clarifying both the cell types that are present, their markers, and their relationships would be important for readers to be able to follow the work. What are Sertoli cells? Or granulosa cells? Or the “supporting cells”? What is the relationship between Supporting cells*

and Sertoli cells? Why would reduction of one affect the other? It would also be helpful to label some of the the structures/cell types in the IFs as well.

As mentioned above, we have included more information on the normal process of sex determination in mammals (lines 43-64 of the revised version of the manuscript).

5. line 90: "Histological analyses on XY KO gonads revealed a complete absence of male-specific structures, such as testicular cords." This data should be shown. I assume by histology the authors are referring to tissue sections which are not included in the manuscript.

We apologize for the potential confusion. The pictures of these tissue sections were in fact included in the manuscript (Fig. 1 e-h). For instance, Fig. 1f shows the histology of the XY *miR-17~92* KO gonads. Our intention was to point that mutant XY gonads lack testis cords similar to those observed in XY control testes (marked by arrows in Fig. 1e) and are histologically identical to the ovaries of XX mice (Figs. 1g and h). We have modified the corresponding sentences in the text to make this observation clearer (lines 114-117 of the revised version of the manuscript).

6. The authors claim that miR-17~92 KO males display complete sex reversal (line 97), with early signs of ovarian development and germ cell meiosis (line 95). Do these animals show oocyte maturation? If so, can you show examples? If not, can this be called complete sex reversal?

Since XY *miR-17~92* KO mice die neonatally, they do not reach the age of puberty, when oocyte maturation process would begin. In any case, the fact a) that no testicular development occurs, b) that at E12.5 the ovarian genetic program is uniformly activated across the gonad, and c) that female meiosis occurs at E14.5 is sufficient to conclude that there is complete sex reversal.

7. Figure 2a: instead of log2-fold changes can you show an unsupervised clustering of the samples based on the genes with disomorphic expression? This will give a better sense of how the KO samples segregate relative to controls.

We believe that the heatmap in Fig. 2a clearly shows that the transcriptomic profiles of the two XX and the XY-KO samples are similar to each other and different from that of the XY controls. As suggested by the reviewer, we also show how they group together in the Extended Data Fig. 2a, which provides a complementary view through a multidimensional scaling (MDS) plot. This plot is derived using unsupervised clustering techniques. It illustrates how the XY *miR-17~92* KO samples cluster together with the two XX samples. If necessary, both figures can be interchanged.

8. Line 128: "Gene Ontology analysis identified several common GO terms between E11.5 mutant and controls, in both XX and XY backgrounds (Extended Data Fig. 2b; Extended Data Tables 10 and 11)". This sentence is confusing and should be re-written. The GO terms are not common between mutants and controls. They are enriched in when looking at genes deregulated between WT and KO animals in both XX and XY backgrounds.

The sentence has been re-written as suggested by the reviewer (lines 156-157 of the revised version of the manuscript).

Line 132: "In particular, we found GO terms related to cell proliferation and molecular pathways involved in its control." Please specify what terms you refer to in the text.

We have added in parentheses the terms referred to (lines 160-161 of the revised version of the manuscript).

9. The authors find that the genital ridge of the mutant gonads contains less proliferative progenitor cells at E11.0. What is the genital ridge and how does it relate to the lower abundance of Sertoli cells at 11.5?

The genital ridge (or gonadal primordium) is the embryonic gonad just after its emergence along the ventromedial surface of the mesonephros. In other words, it is the embryonic gonad in the early stages of development, before sex differentiation. This primordium is the main source for the somatic lineage of the gonad, including Sertoli cells.

The definition of the “genital ridge” term is now expanded in the introduction section (lines 52-53 of the revised version of the manuscript).

10. Line 162: I am very confused on why after characterizing the sex-reversal phenotype in KO animals generated through recombination of a floxed mouse strain the authors have chosen to follow up their studies with embryos derived from tetraploid aggregation following deletion of the cluster in mouse ESC. This strategy allows researchers to generate “fully ESC-derived embryos” by complementing the ESC with tetraploid embryos that can only give rise to extra-embryonic tissues. While a useful strategy to study embryonic development when mutations in the extra-embryonic tissues lead to embryonic lethality, they provide no added benefit to the current study. If anything, they can contribute to additional confounders in the data as the embryonic development will be highly dependent on the ESC quality. The authors should clarify why tetraploid complementation was used for these studies. In addition, they need to provide a full characterization of the ESCs and how the miR-17~92 modification was accomplished. From Figure 2 it seems the authors modified both males and female ESCs. Is the deletion in both lines identical? How many clones per targeting were tested? What are the authors using as controls for these experiments? Embryos generated by tetraploid complementation using parental ESC lines? Or Embryos generated by tetraploid complementation using ESC lines that underwent a “mock” targeting? The authors say “these mutants displayed an identical phenotype to those previously generated via Cre-Lox breeding” but this data also needs to be provided.

Reviewer 1 also raised similar issues (point 1), so the reasons for switching to CRISPR/Cas are already explained above. Basically, this is related to the low yield of mutant embryos obtained through Cre-LoxP breedings, which make scRNA-seq experiments unfeasible. An added benefit of using two alternative methods is that it demonstrates that the phenotype is consistent regardless of the approach employed to generate miR-17~92 KO mutants.

We have now included the corresponding explanation on mESC origin and modifications in the material and methods section (lines 448-460 of the revised version of the manuscript). We have also composed a new supplementary figure to showcase the phenotype of the CRISPR mutants (Extended Data Fig. 3).

From Figure 2 it seems the authors modified both males and female ESCs. Is the deletion in both lines identical? How many clones per targeting were tested? What are the authors using as controls for these experiments? Embryos generated by tetraploid complementation using parental ESC lines? Or Embryos generated by tetraploid complementation using ESC lines that underwent a “mock” targeting?

Actually, the data from Figure 2a-c is derived from embryos resulting from CRE/LoxP breedings. Therefore, all the corresponding XY and XX controls are littermates from the mutant embryos. We only used CRISPR-modified XY mESC cells in tetraploid complementation assays for the single cell experiments (Figure 2d-e) and the reasoning is detailed above (see response to Reviewer 1).

We tested two different mESC clones for the *miR-17~92* CRISPR KO line, both carrying the same deletion and resulting in the sex-reversal phenotype. For the single cell experiment only one of these clones was used. The XY control used for this experiment is derived from tetraploid aggregation of the parental wild type XY G4 -mESC. The XX control is derived from breeding between C57BL/6J males and 129 females, which yield a similar background as the mESC used for the CRISPR experiment. We used this source because XX mESC generally produce a very low yield of embryos in tetraploid aggregation experiment (see response to Reviewer 1 for a more detailed explanation). Despite the fact that XX gonads are derived from breeding, this sample integrates perfectly, and as expected, with the XY gonadal data.

We added this new information in the main text and in the Material and Methods section

The authors say “these mutants displayed an identical phenotype to those previously generated via Cre-Lox breeding” but this data also needs to be provided.

We have composed a new figure for the *miR-17~92* CRISPR KO line displaying the coordinates of CRISPR deletion and the phenotype.

11. Line 168: please clarify in the main text (not just the figure) which embryonic stage was profiled by single-cell sequencing. Also, what tissues were used for this experiment? I assume dissected gonads? But it is not described anywhere.

As requested by the reviewer, we have added now this information in the main text and in the material and methods section.

12. The authors note that Sertoli cells are highly reduced in KO male gonads. Do the authors think this is a consequence of the proliferation defects observed earlier in the genital ridge? The authors say the lack of Sertoli cells indicate a defect in the timing of their differentiation. What are the precursors for Sertoli cells? Do they display defects in proliferation?

At the onset of mammalian sex determination, the testis-determining gene *SRY* is expressed in a precursor cell population known as pre-Sertoli cells. *SRY* expression activates a genetic cascade leading to the differentiation of Sertoli cells, which subsequently orchestrate the differentiation of the bipotential gonad as testis rather than ovary.

In our study, at the sex determination stage, XY *mir-17~92* KO gonads are smaller than XY control ones. However, our scRNA-seq experiments revealed that all cell populations in these gonads are present in similar relative numbers to those in XY controls, except for Sertoli cells. This indicates that a failure of Sertoli cell differentiation rather than a defect in cell proliferation is the underlying cause of the reduced number of Sertoli cells. Consistent with this, we show that the genetic male program is compromised in pre-Sertoli cells (Figure 3).

From the discussion it seems the authors favor a deregulation of the MAPK pathway as a mechanism driving sex reversal. This would certainly fit with roles of the miR-17~92 cluster in other tissues and cell lines and it seems also with sex reversal literature. The manuscript would benefit from exploring this aspect further. Particularly, it would be important to show not only deregulation of genes of this pathway but also that the pathway itself is deregulated in mutants (by IF or western), and to define both the timing of that deregulation and how it relates to the morphological (i.e. lack of Sertoli cells) and molecular (i.e. timing of Sry onset) defects described here, as well as which cell types are affected.

The experiments proposed by the reviewer are interesting, but not particularly straightforward. The MAPK pathway is highly complex and with numerous components, most of which are deregulated in *miR-17~92* mutants, including several that lack any putative target site for these miRNAs. Furthermore, we suspect that no clear differences could be detected between mutant and control gonads. As the reviewer also points out below, this type of regulation by miRNAs involves variations of very modest amplitude in transcript abundance of many different genes, which probably induces in turn modest variations in protein content as well. As such, we believe that the proposed investigation may represent enough work for one (or more) additional paper and feel that this is beyond the scope of our present manuscript.

In the discussion (line 330) the authors write: “This proliferative reduction in mutant gonads appears to affect all progenitor cell types similarly, as their relative numbers do not differ significantly from those of controls, notably in the case of the pre-supporting cell lineage (Fig. 2f).” Can the authors confirm that that is the case (perhaps by co-IF with progenitor cell markers or by FACS)? “Hence, the number of pre-Sertoli cells present at the time of sex determination (E11.5), although reduced, should be sufficient to initiate testis differentiation.” Why? Is it known what is the minimum number of pre-sertoli cells required to support testis differentiation? Are there markers for pre-sertoli cells? If so, can authors quantify and characterized them? Are pre-sertoli cells the same as pre-supporting cells? The authors use both terms to refer to progenitors of Sertoli cells.

Single-cell RNA-seq is perhaps the most powerful and accurate method currently available to identify and quantify cell types in a tissue, as cells are identified based on their transcriptomic signature (hundreds or thousands of different transcripts), which is more defining than a few protein markers that can potentially be detected individually by IF. Therefore, we would argue that IF is not really necessary to confirm the observations derived from scRNAseq data.

As for the other questions, Sertoli and granulosa cells are the supporting cell lineages of the testis and ovaries, respectively. In undifferentiated, bipotential embryonic gonads, these two cell types share the same precursors, the pre-supporting cells, which become pre-Sertoli cells in XY gonads, once they begin to express *Sry*, and pre-granulosa cells in XX ones. In our study, these cell types were identified and quantified with high accuracy in the single-cell RNA-seq experiments. It is known that at least 20% of cells in undifferentiated embryonic gonads have to express *Sry* to trigger testis development [see Burgoyne and Palmer (1993) Cellular basis of sex determination and sex reversal in mammals. In Gonadal Development and Function. Hillier SG ed. New York: Raven Press; 17-29]. In our study (Fig. 2f), the proportion of pre-Sertoli cells is almost identical in both XY controls and XY mutants (22.62 % and 22.74 %, respectively), demonstrating that the lower proliferative rate observed in the mutants does not alter the normal proportion of pre-Sertoli cells, which remains within normal values.

What is the relationship between Sertoli cells and Sry expression? The authors write (line 336): “The lack of Sertoli cells in XY miR-17~92 mutant gonads at the sex determination stage is associated with the heterochronic expression of Sry, which must act within a critical time window to activate Sox9 and trigger the male pathway³⁶”. Do Sertoli cells express Sry? Or is Sry expression required for Sertoli cell differentiation? Please clarify.

Sertoli cells are the only cell type in which *Sry* is expressed and, in fact, it is the onset of *Sry* expression that initiates Sertoli cell differentiation.

We have modified the sentence to make it clearer.

Aside from expanding and clarifying this section of the work, I would also re-structure the

manuscript to make this hypothesis clearer as it seems the most likely explanation for the defects described here.

We have modified the text to express these ideas more clearly (lines 395-396 of the revised version of the manuscript).

13. The authors consistently describe de-regulation of miR-17~92 targets in the manuscript. These should be modified to “predicted” or “putative” targets whenever there is no direct functional validation of targeting.

The reviewer is correct. We have included the terms "predicted" or "putative" in these cases throughout the text.

I would suggest the authors place less emphasis on trying to understand what the direct targets of miR-17~92 driving this phenotype are. It is now well established that miRNAs affect hundreds of genes modestly and that in most cases no single target de-regulation will be able to recapitulate the phenotypes of a miRNA KO. It is highly commendable that the authors attempted to validate Foxl2 as the direct target in this case. However, I would suggest the authors give less emphasis to this section (given that it yields a negative result), moving it down or potentially to a supplementary note. As is, lines 294-307 of the manuscript disrupt the section describing the FOXL2 SOX9 double positive cells. If keeping this section, the same question related to the ESC targeting and controls that I raised regarding the previous tetraploid aggregation experiments need to be included. In addition, the authors should clarify what type of binding site this is (8mer? 7mer-m8? 7mer-A1?). In addition to explaining the source of the control embryos for these experiments, the authors need to include controls of the same developmental ages as the binding-site mutants. For example, Sup Figure 7b shows sequencing data from mutants but not controls.

We have modified the text as suggested by the reviewer. The section on *Foxl2* target site ablation has been moved down to a supplementary note (lines 365-370 of the revised version of the manuscript).

The controls that we used in these experiments had the purpose of testing whether the deletion of the seed region may have any negative effect on the expression of *Foxl2*. For this reason, we could not use XY wildtype gonads as a control (as *Foxl2* is not expressed in that tissue). The control used in this case was the eye of the same mutant embryo, a tissue where *Foxl2* is highly expressed. Thus, we were able to prove that the deletion of a small region of the 3'UTR does not affect the expression of *Foxl2* in a tissue where it is normally transcribed.

The binding site of *Foxl2* is classified into 7mer-m8 category, which includes a seed match extended by a complementary pairing with mRNA nucleotide 8.

14. Controls are also missing in Figure 4a (E11.5 stage).

Controls for the 11.5 dpc stage have been included in Fig. 4A.

Minor comments:

- *Extended Data Figure 1: Mir17hg refers to the human gene. This should be corrected as the figure describes the mouse gene.*

The mouse ortholog gene is also named *Mir17hg* (*Mir17* host gene). See Gene ID: 75957, updated on 11-Sep-2023 in <https://www.ncbi.nlm.nih.gov/gene?b=gene&Cmd=DetailsSearch&Term=75957> and the MGI nomenclature (<https://www.informatics.jax.org/marker/MGI:1923207>).

- *Extended Data Figure 1: the authors show the sequences of the mature miRNAs, but label them as “target sequence”. This may lead readers to think it refers to the sequence of the miRNA target. Label should be changed to “miRNA sequence” to make things clearer.*

The suggested change has been implemented.

- *Extended Data Figure 1: (b) label is missing. In addition, it is unclear what measurements are being plotted in the graphs. What does log intensity value refer to? The figure legend says this data comes from the work of Jamerson and colleagues, but readers should not have to go to that publication to understand the data shown here. Please clarify. In addition, in page 2 lines 68-69 (“Interestingly, miR-17~92 cluster members are expressed in mouse gonads from both sexes during and after the sex determination stage (Extended Data Fig. 1b)”) the authors should add a reference to the Jamerson work when referring to this data.*

The label has been added (thanks for spotting this mistake). As for the meaning of "log intensity values", these are graphs of the "log-transformed, normalized intensity values" from the microarray study by Jameson et al., (2012; doi.org/10.1371/journal.pgen.1002575). We used the miRNA data from these microarrays to study the expression levels of individual miRNAs at three stages of early gonadal differentiation and employed the same nomenclature that the authors used for these types of plots. This has been explained in the legend of the figure (lines 644-645 of the revised version of the manuscript).

- *Line 227: process instead of “process”*

The mistake has been corrected (line 262 of the revised version of the manuscript).

- *Line 242: should maintaining be replaced by “repressing”?*

We believe that it is better to keep the formula "keep ... downregulated" because the term "repress" sounds like a stronger action that could lead to complete inactivation of these genes, which is not the case.

Reviewer 3 (Remarks to the Author):

In mice, expression of the testis-determining gene Sry must reach a threshold for a short period of time to induce differentiation of the testis from the bipotential gonad. The regulatory mechanisms regulating Sry expression have been extensively studied showing that the activation of Sry involves the phosphorylation of GATA4 by the mitogen activated protein kinase (MAPK) pathway. A delay in Sry expression or an expression level that is too low, results in sex reversal. The authors show the potential involvement of a new factor, the miR17-92 in the regulation of Sry in mice. The delay of Sry expression is well documented in the miR17-92 mutants but the causes remain elusive.

This is a potentially very interesting finding. However, it remains to be demonstrated that the phenotype does not result from a global effect of growth retardation but from a specific gonad defect.

If it is not such a defect that leads to the sex reversal phenotype, the authors need to explain why there is a delay in Sry activation.

We thank the reviewer for the positive comments on our findings and the suggestions for improvement. We specifically discuss all these subjects below, as they are referred to in the list of major issues.

Major issues:

The results are based on conditional deletion of miR17-92, and although the sex reversal phenotype is clear, the level of miR17-92 deletion remains to be demonstrated by quantitative PCR experiments. The authors need to show the efficacy of the recombination of the CAG-CRE on miR17-92 (Figure 1). This is a very important point because the entire study and conclusions are based on the impact of miR17-92 deletion.

We agree with the reviewer in that this is an important point to clarify. In this case, we have analyzed our transcriptomic data from mutant and control gonads. These results are shown as a bam sorted file in a modified version of Extended Data Fig. 11 (Extended Data Fig. 10 in the previous version of the manuscript). While we observe a considerable number of reads piling up at the *miR-17-92* locus in controls, the equivalent region in the KO mutants is completely devoid of signal. This demonstrates that the efficiency of CRE recombination is very high, at least in gonadal tissue.

Why did the authors use two different models, an inducible deletion model and a tetraploid model? Would crosses between mutants heterozygous for the knock-out allele not have produced homozygotes? Are homozygous embryos lethal very early and what is the phenotype? This is a source of confusion, and it would be advisable to clarify in each experiment which model was used and why?

We initially crossed *Tg-Cag-Cre* with *mir17-92^{flox/flox}* mice to obtain heterozygotes, which were crossed with each other to generate homozygous KO mice. However, we were unable to detect vaginal plugs in most of these crosses, having a very small number of litters. This is probably due to reduced fertility of *mir17-92^{+/-}* females, as we had previously shown that heterozygous males are fertile (Hurtado et al 2020; <https://doi.org/10.1093/molehr/gaaa027>). Therefore, we backcrossed the heterozygotes with the floxed mice to obtain homozygous mutants. The reasons why we used the tetraploid model for the sc-RNAseq experiments have already been explained above (first question of reviewer 1). Briefly, since these crosses only yield 1/8 of XY KO mice, it would be very difficult for us to collect the gonad pools we needed for the sc-RNAseq experiments and we opted to use the CRISPR/Cas technology coupled with tetraploid aggregation experiments, which yielded isogenic

XY mutant homozygous litters. Of course, we verified that these new mutants exhibit the same phenotype as those obtained with the Cre-LoxP strategy. A similar explanation has been included in the material and methods section (lines 516-530 of the revised version of the manuscript) and a new supplementary figure has been added with pictures of the CRISPR/Cas-derived mutant gonad phenotype (Extended Data Fig. 3).

The embryos die before birth suggesting a pleiotropic effect of conditional deletion of miR17-92 in the embryo or in the placenta. In Figure 1, the kidneys are smaller (this is particularly obvious in the XX mutants). It is important to discriminate whether the reduction in miR17-92 level affects the whole embryo or whether it affects the gonad more specifically. Indeed, a developmental delay could indirectly impact Sry expression.

The authors need either to use a cre recombinase that specifically delete miR17-92 in gonads, or they need to study the development of other organs.

A global effect of embryonic growth retardation has been observed in many studies of gene deletion in which no case of sex reversal was reported [e.g. *Ovca1* (Chen and Behringer, 2004; <http://www.genesdev.org/cgi/doi/10.1101/gad.1162204>.); *MRG15* (Tominaga et al., 2005; <https://doi.org/10.1128/MCB.25.8.2924-2937.2005>); YB-1 (Uchiumi et al., 2006; <https://doi.org/10.1074/jbc.M605948200>); PTB (Shibayama et al., 2009; doi:10.1111/j.1742-4658.2009.07380.x) among others]. Also, the pioneering work by Schmahl and Capel (2003; [https://doi.org/10.1016/S0012-1606\(03\)00122-2](https://doi.org/10.1016/S0012-1606(03)00122-2)) showed that inhibition of proliferation before or after a critical period, that coincides with the initiation of *Sry* expression, led to smaller gonads, but does not block testis differentiation. Moreover, in gonads lacking *Fgf9*, proliferation in the coelomic region of the developing XY gonad is disrupted, but *Sry* transcription is unaffected [Kim et al., (2006; <https://doi.org/10.1371/journal.pbio.0040187>); Schmahl et al., (2004; <https://doi.org/10.1242/dev.01239>)]. To our knowledge, there are only two studies in which reduced proliferation was associated to *Sry* expression disruption. Bogani et al (2009; <https://doi.org/10.1371/journal.pbio.1000196>) showed that loss of MAP3K4 resulted in reduced cell proliferation with *Sry* expression failure. To explain this phenotype, the authors proposed a model in which loss of MAP3K4 renders the coelomic region cells unable to efficiently transduce the FGF9 signalling, causing insufficient provision of pre-Sertoli cells. The authors sentenced: “*Thus, there is no established mechanistic link between prior proliferative defects in the early gonad and subsequent loss of Sry expression*”. Pitetti et al. (2013; <https://doi.org/10.1371/journal.pgen.1003160>) reported that both gonadal cell proliferation and *Sry* upregulation were disrupted in embryos deficient for *Insr:Igflr*. In this case, by using Sf1+ cell sorting, the authors showed that the reduction in the number of somatic progenitors observed in mutant gonads prior to sex determination was not severe enough to cause sex reversal by itself. Similarly, using scRNA-seq, we have found that the pre-Sertoli cell population in *miR17-92* mutants is analogous to that of controls and thus large enough to trigger testis development.

In any case, two facts exclude the possibility that the *Sry* expression delay observed in *miR17-92* mutants is a consequence of general body embryo growth retardation: 1) Ventura et al. (2008; <https://doi.org/10.1016/j.cell.2008.02.019>) generated and described the embryonic development of *mir17-92*^{-/-} mice, finding that they have heart, lung, and B cell differentiation defects. The authors also indicate that a reduction in body size is observed starting at E13.5, but not at earlier stages. 2) Our results are in full agreement with this observation. At stages between E10.5 and E12.5, the key period for gonadal differentiation, our mutant and control embryos were the same size, whereas at E17.5 and later stages, control embryos were noticeably larger than mutant ones. Consistent with this, at E17.5 (the stage shown in Figs. 1a-d), the kidneys of mutants are clearly smaller than those of controls, whereas they look very similar at E14.5. This difference may be due to a specific role of the cluster in kidney development, as the *miR-17~92* cluster has been shown to promote kidney cyst growth in polycystic kidney disease (Patel, 2013). Concluding, the fact that the gonads of mutant embryos are clearly smaller than those of controls, even with similar body size at the time of

sex determination is evidence for a gonad-specific effect of *mir17-92* ablation. To prevent other readers from having the same doubts, this information has been included in the text (lines 384-386 of the revised version of the manuscript).

If the authors find that there is no growth retardation of the embryos, what is the explanation for the delay of Sry expression? The main regulators of Sry is the MAPK pathway including activation by Gadd45g and phosphorylation of GATA4. Gadd45d shows reduced expression in miR17-92 mutant gonads. The labs of A.Greenfield and C.Niehrs have studied the impact of Gadd45g deletion on phosphorylation of p38MAPK and GATA4 thus explaining the delay of Sry expression (Dev Cell 2012 and after). Does the delay in Sry expression result from reduced GATA4 phosphorylation and reduced activation of the MAPK pathway?

It is important to keep in mind that we have deleted a cluster of miRNAs. These molecules have been shown to cause very moderate transcriptional changes across numerous direct downstream targets, all at once. These effects are fundamentally different from those expected from the deletion of a transcription factor or a signaling molecule, as in the studies cited by the reviewer. In this context, to find an explanation for how deletion of *mir-17~92* can lead to sex reversal we focused on the main mechanism of action of miRNAs: their interaction with their putative target genes. Our results are in agreement with those obtained by Han et al. (2015) studying the tail and embryonic heart of *mir-17~92* KO mice. In this study, the authors concluded:

“The last and perhaps most notable conclusion is that, although loss of miR-17~92 or its individual components results in reproducible deregulation of hundreds of genes, the amplitude of this effect is invariably very modest, with the vast majority of genes changing in expression by less than one log₂ (fold change) compared to wild-type embryos. Interestingly, this is not only true for differentially expressed genes that have predicted binding sites for miR-17~92 components in their 3' UTRs—the 'direct' targets—but is also true for genes that are regulated indirectly by miR-17~92 miRNAs. This observation indicates that the modest effects on direct targets are not amplified downstream and has important implications because it suggests that, rather than acting as genetic switches for specific signaling pathways or transcription factors, the miRNAs in the miR-17~92 cluster act as fine-tuners, ensuring that the expression of a large number of genes stays within a very narrow range at critical developmental stages. As a consequence, it is unlikely that the complexity of phenotypes observed in the various mutant mice described here can be assigned to loss of repression of one or a few key targets.”

In *mir-17~92* KO gonads, we found a large number of putative target genes that are preferentially upregulated, and many non-target genes that are deregulated as well. We further show that affected genes belong to gene network/pathways or are involved in processes related to testis development, *Sry* regulation, cell proliferation, WNT signalling, precursor metabolites and energy and MAPK pathway. *Sry* itself is not a target of *miR-17~92*, but misregulation of one or a combination of several of these pathways are probably indirectly responsible for the delay of *Sry* expression observed in mutant gonads. In this context, the complex mechanism of action of miRNAs makes it very difficult to assign individual components of a gene pathway a main role during regulation. Specifically, in the GADD45G/p38 MAPK/GATA4/SRY axis, we found that both *Gadd45g* and *Sry* were downregulated in mutant gonads at the time of sex determination. We also identified other deregulated genes belonging to this pathway which were not suspect of having a possible role in mammalian sex determination [see Ext. Data 9a; Gierl (2012; <https://doi.org/10.1016/j.devcel.2012.09.014>) and Warr (2012; <https://doi.org/10.1016/j.devcel.2012.09.016>)]. It is probably that we do not yet know all the interactions at the molecular level that govern this axis in this developmental process. As we have mentioned, the magnitude of gene deregulation in mutant gonads is very modest (less than 2-fold in most cases), so we propose that *mir-17~92* fine-tunes the expression of target components of this pathway, in addition to others, to produce accurate

expression timing of *Sry*. This is a novel level of regulation that was thought to influence neither mammalian sex determination nor *Sry* regulation.

Our study represents the first evidence that miRNAs play an essential role in mammalian sex determination. As such, we are aware that there are still open questions (such as a potential involvement of GATA phosphorylation and others), as pointed out by the reviewer. These are of course questions that we would be interested to answer in the future, but that fall beyond the scope of the present manuscript.

Another important step in testis development is the epigenetic status. The authors show that abundant double positive cells for SOX9 and FOXL2 are detected in the mutant gonads. Similar observations have been reported in the Cbx2^{-/-}, a member of the PRC1 complex involved in epigenetic regulation. <https://doi.org/10.1371/journal.pgen.1007895>

The regulation of gonad differentiation by miR17-92 is potentially a very interesting discovery, but it remains to be clarified how this miR is involved in sex determination (the MAPK pathway, histone methylation, both, another mechanism?).

The reviewer suggests several interesting experimental analyses that would help to better understand the molecular mechanisms connecting *miR17-92* to *Syr* regulation. These experiments, as well as others such as identifying which particular miRNA(s) of the cluster are responsible for sex reversal are worthwhile, of course, but represent a volume of additional work that would be sufficient to warrant a separate article. Our study focuses on the effects of *miR-17~92* ablation: we have shown that bipotential gonad proliferation is compromised, that *Sry* expression is delayed, and that male-to-female sex reversal subsequently occurs. We have also identified the major gene networks affected, including those directly involved in *Sry* regulation. Indeed, our results strongly suggest that the delay of *Sry* expression is caused by deregulation of the MAPK pathway, including down-regulation of *Gadd45g*. We believe that studying also protein phosphorylation, chromatin remodeling and DNA methylation status of specific genes is very interesting but beyond the scope of this article (also in agreement with the opinion of reviewer 2).

Additional comments

Line 24 whereas its absence in XX embryos leads to ovarian differentiation.

*Authors need to be more cautious in describing previous results. Some of the results obtained in mice may not be true in all mammals, and in humans. In humans, mutations in the SRY gene results in 46,XY complete gonadal dysgenesis, which is not ovarian differentiation. I suggest that the authors replace mammals (line 22) by mice and SRY by *Sry* or ovarian differentiation by complete gonadal dysgenesis.*

Same comment lines 40 to 44.

We agree with the reviewer that care should be taken in the abstract and the introduction to make it clear when we are talking about studies performed in humans or in mice because, as the reviewer says, "*some of the results obtained in mice may not be true in all mammals, and in humans*". However, after carefully reading lines 24 and 40-44 of the previous version of the manuscript, we conclude that our text is not incorrect. We state that the absence of *SRY* in XX gonads results in ovarian development and this is true for all mammals that have *SRY*. We are talking about normal sex determination in XX embryos (*SRY* is normally absent in XX embryonic gonads and that is why they develop as ovaries), not about cases of *SRY* mutations in 46XY humans. Later in text we have explicitly indicated that we are talking about result obtained in mice (line 61 of the revised version of the manuscript).

Line45. Reference 8 is not relevant. The downregulation of WNT signaling in XY gonads has been clarified and involves ZNRF3 as evidenced by Harris/Greenfield (2018).

The reviewer is right. The reference 8 has been removed

Line 50 remains instead of remain

We thank the reviewer for spotting this spelling mistake. We have corrected it (line 71 of the revised version of the manuscript).

Line 59. Huang and Yao have used the Sf1-cre (Bingham et al, 2006) to delete Dicer 1. Manuylov/Tevosian (2011) have shown that the Sf1Cre recombined inefficiently in coelomic epithelial cells and is not very efficient around E11.5. This might explain the absence of an obvious phenotype when Dicer is deleted with this cre recombinase.

We thank the reviewer for this comment. We have included this additional explanation for the lack of phenotypic effects after *Dicer1* ablation experiments in the text (lines 81-82 of the revised version of the manuscript).

Line 84 Although the cre recombinase used by the authors is described in Mat and Met, it should be added in the results section. The gene excision data is confusing (The legend of extended Data Figure 10 should be more detailed). The authors' current attempts to clarify this point have to be better shown. This is important because excision efficiency is key to the authors' findings. The authors need to quantify the level of miR17-92 deletion using quantitative RT-PCR as previously described in Hurtado/Barrionuevo (2018).

The Cre recombinase used has also been indicated in the results section (line 109 of the revised version of the manuscript). To show excision efficiency in *mir-17~90* KO testis we opted to employ a representation of the aligned reads from the RNA-seq data. In the Extended Data Figure 10 (previous version of the manuscript) we displayed a representation of a bigWig file, as is sometimes recommended. (<https://genome.ucsc.edu/goldenPath/help/bigWig.html>). However, we agree with the reviewer that it can be confusing for those who are not familiar with this type of graphics. In addition, we also agree that the description of the figure was insufficiently detailed. Thus, we have plotted instead the transcriptomic data of .bam (sorted) files, and wrote a more descriptive legend. In these plots, each small segment represents a mapped read. As we can see in the new Ext. Data Fig. 11, the mutants completely lack transcript reads mapping to the *miR-17~92* cluster, proving the absolute efficiency of our targeting strategy. In the figure, we display 2 controls and 2 mutants at E11.5, but similar results are obtained for the other controls and mutants at both E11.5 and E12.5.

Regarding quantification of *miR17-92* levels, in the Hurtado/Barrionuevo (2018) paper we performed bulk RNA-seq of adult testis, as we conditionally deleted *miR-17~92* in Sertoli cells. However, as Sertoli cells account for a small fraction of the number cells in the adult testis, we could not use the transcriptomic data to assess deletion efficiency. Thus, we performed RT-PCR in embryonic testes. However, in this study, we have deleted the miRNA cluster in the whole embryo, and thus, we can use the transcriptomic data which provides much more accurate and detailed information than a RT-PCR.

Can the authors comment why did they not use the mating between miR17-92+/- heterozygous mice to get homozygous miR17-92-/- embryos?

This question has already been answered above (see the second paragraph of the "Major issues"

section of reviewer 3).

Figure 1: The development of the embryos seems to be impaired by genetic deletion of miR17-92. The impact of miR17-92 deletion in the whole embryos/other tissues of the embryos need to be documented (see my comments above).

Those results were already published by Ventura et al., (2008; <https://doi.org/10.1016/j.cell.2008.02.019>) and we have mention them above (answer to the third paragraph of the "Major issues" section of reviewer 3).

The official name of OCT4 is POU5F1.

The reviewer is correct, we have changed it (lines 134-135 of the revised version of the manuscript).

Line 97: XY mice undergo complete primary sex reversal.

We have deleted “and develop as phenotypic females.” (line 124 of the revised version of the manuscript)

Line 110 Contrarily, all germ cells of the other three genotypes had entered meiosis. How did the authors show that all germ cells have entered meiosis?

We know that "all" germ cells have entered meiosis because there are none left expressing *POU5F1*.

Figure 2

Figure2c. WT1 is expressed in the gonad and the mesonephros (Liu/Yao 2016, Sasaki/Saitou 2021, Mayere/Nef 2022). In these images, WT1 immunostaining is unexpected because it seems to be restricted to the coelomic epithelium in XY control at E11.0 and not expressed in the mesonephros at E11.0 and after. Is there an effect of genetic deletion of miR17-92 on WT1? This needs to be clarified.

Figure 2c shows an enlarged section of a larger original micrograph, which shows the gonad almost exclusively. In addition, we merged the green (WT1) and red channels (Ki67) to better show the proliferation of WT1+ cells. Because of this, WT1 expression in the mesonephros that remains in the figure is masked. As mentioned by the reviewer, WT1 is expressed in both the gonad and the mesonephros of control and mutant embryos, as seen in the original green channel micrographs:

Figure 2 d. For clustering and identification of populations, cells were coloured by genotypes (d). The authors show only a merged image, which is difficult to evaluate. The authors should show images for each color/genotype separately together with merged images so that the readers can examine each genotype appropriately.

We have modified Figure 2d as suggested by the reviewer.

Figure 2f. The pre-granulosa cells are not represented whereas immunostaining of FOXL2-positive cells show that these cells are present around E11.75 as shown in Figure 4b. Are they present or absent in the single cell RNA seq analysis? If they are absent, can the authors comment on the discrepancy between Figure 2 and 4?

At these early stages, both pre-granulosa and pre-Sertoli cells are clustered together in the pre-supporting cell lineage.

All along the text, the authors mainly used the term of “granulosa cell” to design the female supporting cell lineage. Are they sure this term is appropriate before follicle formation occurs? Pre-granulosa cell seems commonly used before follicle formation.

This is a very interesting observation. We have substituted “granulosa cell” with “pre-granulosa cell” in stages previous to follicle formation (lines 63, and 313 of the revised version of the manuscript).

Figure 3: the delay of SRY is convincing. It doesn't have to be backed up by statistics. For small sample sizes ($n=3$) descriptive statistics can be not appropriate. Please check the policy of the journal.

We used our *Sry* and *Sox9* transcriptomic data to support the IF observations and performed statistical analyses using EdgeR, which has statistical robustness even with only three samples in transcriptomic analyses. This approach complies with the policy of the journal.

Figure 4: While the expression of FOXL2 and SRY can be observed when SRY is delayed and in turn SOX9, the double positive cells FOXL2/SOX9 are usually rare. (Garcia-Moreno/Capel 2019) showed that cells in the *Cbx2*^{-/-} XY gonad express SOX9 and FOXL2 simultaneously, with a reduction in SOX9-positive cells over time. Their results suggest that the initiation of the testis program occurs, but the stabilization phase, in which the female pathway is silenced by epigenetic chromatin regulation, is disrupted in the absence of CBX2. It is thus conceivable that the phenotype

due to reduced miR17-92 expression could modify the epigenetic state of chromatin. Is it a cause or a consequence?

This is a very interesting issue, for which we (yet) do not have an answer. As mentioned above, we believe that studies on the epigenetic status of chromatin states could be the subject of future research but is beyond the scope of the current study.

Figure 4b. How were the cell percentages quantified, and how many cells were counted?

For FOXL2/SOX9 cell counting, we used 3 XY mutant gonads of every stage (22-23ts and 27-29ts). In every gonad, we counted the positive cells of at least 2 20× micrographs as follows: 1) we use the red channel to count the total number of SOX9+ cells (Sox9_tot), 2) we used the green channel to count the total number of FOXL2+ cells (Foxl2_tot), 3) we used the merged channel (yellow) to count the number of double SOX9/FOXL2+ cells (Double_tot). Then we calculate the number of cells which express only 1 gene: Sox9_only = Sox9_tot – Double_tot; Foxl2_only = Foxl2_tot – Double_tot. Finally, in Fig 4b we represented Sox9_only, Double_tot and Foxl2_only. The total number of cells at 22-23ts were Sox9_tot = 216, Foxl2_tot = 250, Double_tot = 188; and at 27-29ts Sox9_tot = 208, Foxl2_tot = 614, Double_tot = 190.

REVIEWER COMMENTS

Reviewer #1 (Remarks to the Author):

The authors have satisfactorily addressed the questions I had raised in the first round of review.

Reviewer #2 (Remarks to the Author):

I commend the authors on the revisions they have performed for this manuscript. They have addressed all my request and I have no additional concerns. The manuscript is clear and I think this is an important contribution to the miRNA field.

Reviewer #3 (Remarks to the Author):

The authors have answered most of my comments but not all, or they have only been addressed partly.

1-The gonads are hypoplastic in the miR17-92 KO embryos. When such a phenotype is described, it is mandatory to compare the whole embryo with a wildtype. Indeed, the pioneer work of Schmahl and Capel 2003 revealed an 8-h window, between approximately 10.8 and 11.2 dpc using cell proliferation inhibitors, that was critical for establishing expression of pro-testis genes and testis cord formation. Inhibition of cell proliferation either side of this window did not block testis development. This suggests that proliferation is an important step of testis development. The proliferation at this early stage is reduced in miR17-92 KO in both sexes. This is an important result and it remains to clarify whether it is associated with a general defect as suggested by the reduced proliferation in both sexes or a gonad specific proliferation defect. This can be addressed by a comparison of size of the embryos or maybe, the authors can use the data from Ventura et al (2008). The comment added in discussion comes too late, it must be described when the hypoplastic gonadal phenotype is described.

2- As asked in the first review, the authors need to clarify which model has been used in each figure. Indeed, XY 92-27 KO describes Tg-Cag-Cre with mir17-92flox/flox mice and "CRISPR/tetraploid" embryos, it would be more accurate that the two models have different names. It is very difficult to know which mutation the authors used in each experiment. This information should be very easy to find, and I propose that the model(s) used should be clearly identified in each figure at least in each legend of figure. Every reader should be able to understand the results by studying the figures alone.

Lastly, the results are based on conditional deletion of miR17-92, and although the sex reversal phenotype is clear, the level of miR17-92 deletion remains to be demonstrated. The authors need to show the efficacy of the recombination of the CAG-CRE on miR17-92 in Figure 1 and not in an extended data Figure 10 or 11 in the present version (if the data produced in extended fig 10/11 come from Tg-Cag-Cre with mir17-92flox/flox embryos). The extended data Figure 11 (new version) must be added in Figure1.

Rebuttal letter 2. Manuscript NCOMMS-23-27381-T

REVIEWER COMMENTS

Reviewer 1 (Remarks to the Author):

The authors have satisfactorily addressed the questions I had raised in the first round of review.

We thank the reviewer for this positive comment.

Reviewer 2 (Remarks to the Author):

I commend the authors on the revisions they have performed for this manuscript. They have addressed all my request and I have no additional concerns. The manuscript is clear and I think this is an important contribution to the miRNA field.

We thank the reviewer for the favourable comments.

Reviewer 3 (Remarks to the Author):

The authors have answered most of my comments but not all, or they have only been addressed partly.

We appreciate the reviewer's new comments and suggestions for improvement.

1-The gonads are hypoplastic in the miR17-92 KO embryos. When such a phenotype is described, it is mandatory to compare the whole embryo with a wildtype. Indeed, the pioneer work of Schmahl and Capel 2003 revealed an 8-h window, between approximately 10.8 and 11.2 dpc using cell proliferation inhibitors, that was critical for establishing expression of pro-testis genes and testis cord formation. Inhibition of cell proliferation either side of this window did not block testis development. This suggests that proliferation is an important step of testis development. The proliferation at this early stage is reduced in miR17-92 KO in both sexes. This is an important result and it remains to clarify whether it is associated with a general defect as suggested by the reduced proliferation in both sexes or a gonad specific proliferation defect. This can be addressed by a comparison of size of the embryos or maybe, the authors can use the data from Ventura et al (2008). The comment added in discussion comes too late, it must be described when the hypoplastic gonadal phenotype is described.

As mentioned previously, our observations on the size of mutant embryos during embryonic development are similar to those described by Ventura et al. 2008 (ref. 23 of the manuscript). Being mindful of this issue, we took pictures of every litters we analysed to facilitate potential measurements. As suggested by the reviewer, we have measured the body size (crown-rump length, CRL) of the embryos and generated a new panel in the Extended Data Figure 3, (3c) including a photograph of a litter containing two mutant and one control embryos, and a CRL comparison between controls and mutants at E11.5, showing that there is no statistically significant difference in the body size (see also lines 167-169 of the revised version of the manuscript). For review purposes, we also provide here some pictures of other litters at E11.5, E14.0-E14.5 and E17.0-E17.5, where the reviewers can see that, at E11.5, the gross morphology and size of our mutant embryos is similar to those of the controls, whereas they are smaller at E14.0 and later stages.

2- As asked in the first review, the authors need to clarify which model has been used in each figure. Indeed, XY 92-27 KO describes *Tg-Cag-Cre* with *mir17-92^{flox/flox}* mice and “CRISPR/tetraploid” embryos, it would be more accurate that the two models have different names. It is very difficult to know which mutation the authors used in each experiment. This information should be very easy to find, and I propose that the model(s) used should be clearly identified in each figure at least in each legend of figure. Every reader should be able to understand the results by studying the figures alone.

We apologise for not making this clear in the first review. In agreement with the reviewer, information on the origin of each mutant analyzed here should be provided, as other readers may have similar doubts. Thus, following the reviewer's recommendation, we have added, at the end of the figure legend of every figure (including the supplementary ones), a sentence indicating the method employed to generate the mutants used in each experiment.

Lastly, the results are based on conditional deletion of miR17-92, and although the sex reversal phenotype is clear, the level of miR17-92 deletion remains to be demonstrated. The authors need to show the efficacy of the recombination of the CAG-CRE on miR17-92 in Figure 1 and not in an extended data Figure 10 or 11 in the present version (if the data produced in extended fig 10/11 come from Tg-Cag-Cre with mir17-92flox/flox embryos). The extended data Figure 11 (new version) must be added in Figure1.

We agree with the reviewer that demonstrating the efficiency of the CAG-CRE-induced recombination is important for the reliability of our work. To maintain consistency in the topics covered in each figure, we have preferred to include this transcriptomic information in Figure 2, which contains information on bulk RNA-seq, instead of Figure 1, where the mutant phenotype is described. As Figure 2 was already very large, we have divided it into two. The new Figure 2a now shows a representation of the frequencies of gonadal transcript reads mapped to the *Mir17hg* locus, demonstrating that, in contrast to control gonads, no reads are observed in the *miR-17~92* region of the mutant genome. This clearly shows that this region was effectively deleted (see lines 140-145 of the revised version of the manuscript). Furthermore, we have kept the previous Extended Data Figure 11 (Extended Data Figure 2 in the new version of the manuscript), which shows the Integrative Genome Browser plot depicting RNA-seq reads from E11.5 gonads mapped to the *Mir17hg* locus. We have maintained this graph as an Extended Data Figure due to its large size, which makes it difficult to incorporate into a main figure. Thus, the new Figure 3 exclusively contains information on single-cell RNA-seq. This has forced us to renumber most of the manuscript figures.